# Asymmetric Perturbation in Solving Bilinear Saddle-Point Optimization

**Kenshi Abe** [1]   **Mitsuki Sakamoto** [1]   **Kaito Ariu** [1]   **Atsushi Iwasaki** [2]

## Abstract

This paper proposes *asymmetric perturbation*, where only one player's payoff function is perturbed, for solving bilinear saddle-point optimization problems, commonly arising in minimax problems, game theory, and constrained optimization. Symmetric perturbation is known to require decreasing its strength to ensure convergence to a solution, i.e., an equilibrium in the original game, resulting in a slower rate. First, with asymmetric perturbation, we show that, for a sufficiently small perturbation strength, the equilibrium strategy of the asymmetrically perturbed game coincides with an equilibrium strategy of the original unperturbed game. Second, building on this coincidence, we construct a learning algorithm with a linear last-iterate convergence rate. Third, motivated by the fact that the coincidence relies on the perturbation strength being sufficiently small, we also provide a parameter-free variant, retaining the linear rate. Finally, we empirically demonstrate fast convergence toward equilibria in both normal-form and extensive-form games.

## 1. Introduction

This paper proposes an asymmetric perturbation technique for solving saddle-point optimization problems, commonly arising in minimax problems, game theory, and constrained optimization. Over the past decade, no-regret learning algorithms have been extensively studied for computing (approximate) solutions or equilibria. When each player minimizes regret, the time-averaged strategies approximate Nash equilibria in two-player zero-sum games; that is, *average-iterate convergence* is guaranteed. However, the actual sequence of strategies does not necessarily converge and can cycle or diverge even in simple bilinear cases (Mertikopoulos et al., 2018; Bailey & Piliouras, 2018; Cheung & Piliouras, 2019).

This is problematic, especially in large-scale games with neural network policies, since averaging requires storing a separate model at every iteration.

This motivates the study of *last-iterate convergence*, a stronger notion than average-iterate convergence, in which the strategies themselves converge to an equilibrium. One successful approach is to use *optimistic* learning algorithms, which essentially incorporate a one-step optimistic prediction that the environment will behave similarly in the next step. This idea has led to several effective algorithms, including Extra-Gradient methods (EG) (Liang & Stokes, 2019; Mokhtari et al., 2020), Optimistic Gradient Descent Ascent (OGDA) (Daskalakis & Panageas, 2019; Gidel et al., 2019; Mertikopoulos & Zhou, 2019), and Optimistic Multiplicative Weights Update (OMWU) (Daskalakis & Panageas, 2019; Lei et al., 2021). However, in large-scale settings where the gradient must be estimated from data or simulation, these algorithms can lose the last-iterate convergence property. For example, Abe et al. (2022) reports empirical non-convergence behavior under bandit feedback.

Alternatively, perturbing the payoffs with strongly convex penalties (Facchinei & Pang, 2003) has long been recognized as an effective technique for achieving last-iterate convergence (Koshal et al., 2010; Tatarenko & Kamgarpour, 2019). This line of work has also shown strong performance in practical settings, including learning in large-scale games (Bakhtin et al., 2023) and fine-tuning large language models via preference optimization (Ye et al., 2024), often in place of optimistic algorithms. In prior work, the perturbation is almost always applied *symmetrically*, meaning both players' payoff functions are perturbed by a strongly convex penalty. A known limitation is that, for any fixed perturbation strength $\mu > 0$, the equilibrium of the symmetrically perturbed game generally remains only an approximation of an equilibrium of the original game, and the deviation scales with the strength of the perturbation (Liu et al., 2023; Abe et al., 2024). Consequently, recovering equilibria of the original game typically requires tuning $\mu$ carefully (e.g., as a function of the iteration budget) or employing a continuation-type procedure, which introduces a nontrivial trade-off between accuracy and convergence speed.

To avoid these restrictions, we develop an *asymmetric per-*

---

[1]CyberAgent, Tokyo, Japan [2]University of Electro-Communications, Tokyo, Japan. Correspondence to: Kenshi Abe <abekenshi1224@gmail.com>.

*Proceedings of the 43rd International Conference on Machine Learning*, Seoul, South Korea. PMLR 306, 2026. Copyright 2026 by the author(s).

*turbation* approach, in which only one player's payoff function is perturbed while the other remains unperturbed. This simple modification yields a qualitatively different outcome. For any sufficiently small perturbation strength within a broad and practical range, the equilibrium strategy of the perturbed game coincides with that of the original game (see Corollary 3.2). Intuitively, leaving player $y$ unperturbed preserves the linearity of player $x$'s original objective, so adding a strongly convex perturbation does not significantly shift the solution (see Figure 2). Consequently, solving the asymmetrically perturbed game suffices to recover an equilibrium strategy of the original game.

Furthermore, to demonstrate the effectiveness of our findings, we first incorporate the asymmetric perturbation technique into a gradient-based learning algorithm for bilinear games[1], and show that it converges to a saddle point at a linear (exponentially fast) rate (see Theorem 4.1 and Corollary 4.2). This rate is provably faster than an $\tilde{\mathcal{O}}(1/t)$[2] rate for existing symmetric perturbation-based approaches under the same bilinear setting (Liu et al., 2023). Nevertheless, recovering an equilibrium of the original game in this way still relies on choosing a sufficiently small perturbation strength, although larger choices only lead to a bounded deviation controlled by the perturbation strength. To overcome this, we further provide a parameter-free variant that leverages the same invariance phenomenon to retain a linear last-iterate convergence rate. Second, we apply our asymmetric perturbation to gradient-based learning methods for extensive-form games using a dilated regularizer (Hoda et al., 2010), a standard choice for learning over sequence-form strategy spaces (von Stengel, 1996). While our analysis focuses on bilinear games, the structural insight behind the asymmetric perturbation may extend beyond this setting, including two-player zero-sum Markov games, and serves as a bridge to the design of new perturbation-based learning algorithms.

## 2. Preliminaries

**Bilinear saddle-point optimization problems.** In this study, we focus on the following bilinear saddle-point problem:

$$\min_{x\in\mathcal{X}}\max_{y\in\mathcal{Y}} x^\top Ay, \qquad (1)$$

where $\mathcal{X} \subseteq \mathbb{R}^m$ (resp. $\mathcal{Y} \subseteq \mathbb{R}^n$) represents the $m$-dimensional (resp. $n$-dimensional) convex strategy space for player $x$ (resp. player $y$), and $A \in \mathbb{R}^{m \times n}$ is a game matrix. We assume that $\mathcal{X}$ and $\mathcal{Y}$ are polytopes. We refer to the function $x^\top Ay$ as the *payoff function*, and write

---

[1]An implementation of the method is available at https://github.com/CyberAgentAILab/asymmetrically-perturbed-gda

[2]We use $\tilde{\mathcal{O}}$ to denote a Landau notation that disregards a polylogarithmic factor.

$z := (x, y) \in \mathcal{Z} := \mathcal{X} \times \mathcal{Y}$ as the *strategy profile*. This formulation includes many well-studied classes of games, such as two-player normal-form games and extensive-form games with perfect recall.

**Nash equilibrium.** This study aims to compute a minimax or maximin strategy in the optimization problem (1). Let $\mathcal{X}^* := \arg\min_{x\in\mathcal{X}} \max_{y\in\mathcal{Y}} x^\top Ay$ denote the set of minimax strategies, and let $\mathcal{Y}^* := \arg\max_{y\in\mathcal{Y}} \min_{x\in\mathcal{X}} x^\top Ay$ denote the set of maximin strategies. It is well-known that any strategy profile $(x^*, y^*) \in \mathcal{X}^* \times \mathcal{Y}^*$ is a *Nash equilibrium*, which satisfies the following condition:

$$\forall(x,y)\in\mathcal{X}\times\mathcal{Y},\ (x^*)^\top Ay \le (x^*)^\top Ay^* \le x^\top Ay^*.$$

Based on the minimax theorem (v. Neumann, 1928), every equilibrium $(x^*, y^*) \in \mathcal{X}^* \times \mathcal{Y}^*$ attains the identical value, denoted as $v^*$, which can be expressed as:

$$v^* := \min_{x\in\mathcal{X}}\max_{y\in\mathcal{Y}} x^\top Ay = \max_{y\in\mathcal{Y}}\min_{x\in\mathcal{X}} x^\top Ay.$$

We refer to $v^*$ as the *game value*. To quantify the proximity to equilibrium for a given strategy profile $(x, y)$, we use *NashConv*, which is defined as follows:

$$\mathrm{NashConv}(x,y) = \max_{\tilde{y}\in\mathcal{Y}} x^\top A\tilde{y} - \min_{\tilde{x}\in\mathcal{X}}(\tilde{x})^\top Ay.$$

**Symmetric perturbation.** *Payoff perturbation* is an extensively studied technique for solving games (Facchinei & Pang, 2003; Liu et al., 2023). In this approach, the payoff functions of all players are perturbed by a strongly convex function $\psi$. For example, in bilinear games, instead of solving the original game in (1), we solve the following perturbed game:

$$\min_{x\in\mathcal{X}}\max_{y\in\mathcal{Y}}\left\{x^\top Ay + \mu\psi(x) - \mu\psi(y)\right\},$$

where $\mu \in (0, \infty)$ is the *perturbation strength*. Since the perturbation is applied to both players' payoff functions, we refer to this perturbed game as a *symmetrically* perturbed game. In this study, we specifically focus on the standard case, where the perturbation payoff function $\psi$ is given by the squared $\ell^2$-norm, i.e., $\psi(x) = \frac{1}{2}\|x\|^2$:

$$\min_{x\in\mathcal{X}}\max_{y\in\mathcal{Y}}\left\{x^\top Ay + \frac{\mu}{2}\|x\|^2 - \frac{\mu}{2}\|y\|^2\right\}. \qquad (2)$$

Let $x^\mu$ (resp. $y^\mu$) denote the minimax (resp. maximin) strategy in the symmetrically perturbed game (2) [3], which can be solved at a linear rate (Cen et al., 2021; 2023; Pattathil

---

[3]The minimax strategy is uniquely determined because symmetrically perturbed games satisfy the strongly convex–strongly concave property.

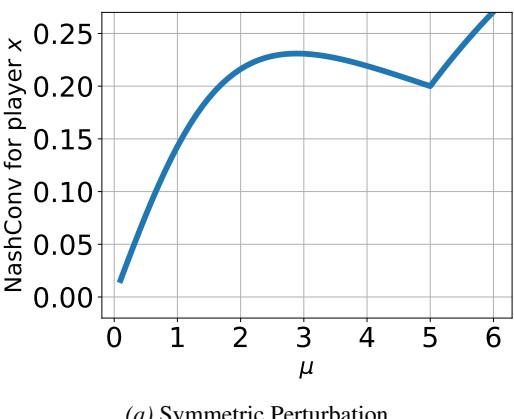

*(a)* Symmetric Perturbation

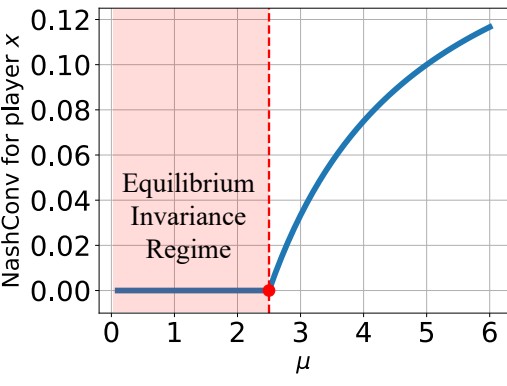

*(b)* Asymmetric Perturbation

*Figure 1.* The proximity of $x^\mu$ to $x^*$ under the symmetric perturbation and the asymmetric perturbation with varying $\mu$. The game matrix $A$ is given by $[[0, 1, -3], [-1, 0, 1], [3, -1, 0]]$. The proximity to the minimax strategy is measured by the value of $\max_{y \in \mathcal{Y}}(x^\mu)^\top A y - v^*$.

et al., 2023; Sokota et al., 2023). It is known that the solution $(x^\mu, y^\mu)$ is only an approximation of an equilibrium of the original game, with an error bounded by $\mathcal{O}(\mu)$ (Liu et al., 2023; Abe et al., 2024). Consequently, typical perturbation-based methods must employ a decreasing schedule for $\mu$, or use an extremely small fixed $\mu$ tuned to the number of iterations $T$ (e.g., $\mu = \mathcal{O}(1/T)$), which requires careful hyperparameter tuning (Tatarenko & Kamgarpour, 2019; Bernasconi et al., 2024; Cai et al., 2023). See Figure 1a for a biased Rock–Paper–Scissors game where the perturbed solution differs from the original equilibrium. We provide a rigorous justification for this phenomenon in Appendix B, showing that for this instance, $(x^\mu, y^\mu)$ fails to coincide with any equilibrium of the original game for every fixed $\mu > 0$.

**Additional notation.** For a closed convex set $\mathcal{A} \subseteq \mathbb{R}^d$, we let $\Pi_\mathcal{A}(a) := \arg\min_{a' \in \mathcal{A}} \|a - a'\|$ denote the Euclidean projection operator onto $\mathcal{A}$, and $\mathrm{dist}(a, \mathcal{A}) := \|a - \Pi_\mathcal{A}(a)\|$ denote the distance from a point $a$ to $\mathcal{A}$. We denote the $d$-dimensional probability simplex by $\Delta^d = \{p \in [0,1]^d \mid \sum_{j=1}^d p_j = 1\}$, and write $\mathbf{1}_d, \mathbf{0}_d$ for the $d$-dimensional all-ones and all-zeros vectors.

## 3. Asymmetric Payoff Perturbation

In this section, we explain our novel technique of asymmetric payoff perturbation. We demonstrate that a seemingly minor structural change—perturbing only one player's payoff—can yield a dramatically different outcome: the solution of the perturbed game exactly matches an equilibrium strategy $x^*$ in many cases.

### 3.1. Asymmetric Payoff Perturbation

Instead of incorporating the perturbation into both players' payoff functions, we consider the case where only player

$x$'s payoff function is perturbed:

$$\min_{x \in \mathcal{X}} \max_{y \in \mathcal{Y}} \left\{ x^\top A y + \frac{\mu}{2} \|x\|^2 \right\}. \quad (3)$$

The procedure we are going to describe in Theorem 3.1 focuses on computing the minimax strategy $x^*$, rather than the maximin strategy $y^*$. To compute $y^*$, we simply solve the corresponding maximin problem for player $y$:

$$\max_{y \in \mathcal{Y}} \min_{x \in \mathcal{X}} \left\{ x^\top A y - \frac{\mu}{2} \|y\|^2 \right\}.$$

The same reasoning applies to this perturbed maximin optimization problem. Thus, hereafter, we primarily focus on the perturbed game (3) from the perspective of player $x$.

Since the function $\max_{y \in \mathcal{Y}} x^\top A y$ is convex with respect to $x$ (Boyd & Vandenberghe, 2004), the perturbed objective $\max_{y \in \mathcal{Y}} x^\top A y + \frac{\mu}{2} \|x\|^2$ is $\mu$-strongly convex. Therefore, the minimax strategy for the perturbed game (3) is unique. We denote it by $x^\mu$ and denote the set of maximin strategies in (3) by $\mathcal{Y}^\mu$. Since both the minimax and maximin strategies constitute a Nash equilibrium of the perturbed game, the pair $(x^\mu, y^\mu)$ with $y^\mu \in \mathcal{Y}^\mu$ satisfies the following conditions: for all $\tilde{y}^\mu \in \mathcal{Y}^\mu$ and $x \in \mathcal{X}$,

$$(x^\mu)^\top A \tilde{y}^\mu + \frac{\mu}{2} \|x^\mu\|^2 \le x^\top A \tilde{y}^\mu + \frac{\mu}{2} \|x\|^2, \quad (4)$$

and for all $y \in \mathcal{Y}$,

$$(x^\mu)^\top A y^\mu \ge (x^\mu)^\top A y. \quad (5)$$

### 3.2. Equilibrium Invariance under the Asymmetric Perturbation

In this section, we discuss the properties of the minimax strategies for asymmetrically perturbed games. Our main result is the following quantitative bound on the deviation of $x^\mu$ from the original minimax strategy set $\mathcal{X}^*$, controlled by the perturbation strength $\mu$:

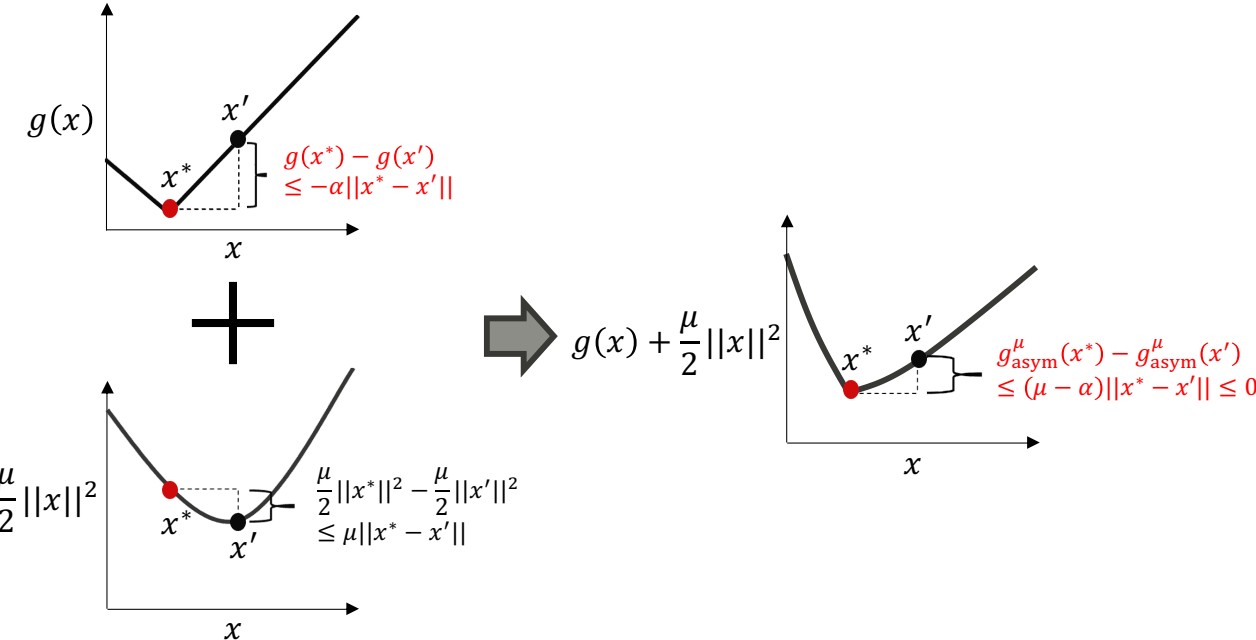

*Figure 2.* The landscape of the objective function for player $x$ in asymmetrically perturbed games. The functions $g(x)$ and $g^\mu_{\mathrm{asym}}(x)$ are defined as $g(x) := \max\limits_{y \in \mathcal{Y}} x^\top A y$ and $g^\mu_{\mathrm{asym}}(x) := g(x) + \frac{\mu}{2} \|x\|^2$, respectively.

**Theorem 3.1.** *For any $\mu > 0$, the minimax strategy $x^\mu$ of the asymmetrically perturbed game* (3) *satisfies:*

$$\operatorname{dist}(x^\mu, \mathcal{X}^*) \leq 2 \max \left\{ 0, \max_{x \in \mathcal{X}} \|x\| - \frac{\alpha}{\mu} \right\},$$

*where $\alpha > 0$ is a constant depending only on the original game* (1)*, given in* (6)*.*

A particularly noteworthy consequence is that the right-hand side of the bound is exactly zero whenever $\mu \leq \alpha / \max_{x \in \mathcal{X}} \|x\|$. In that regime, $x^\mu$ does not just approximate but exactly recovers a minimax strategy of the original game (1):

**Corollary 3.2.** *Assume that the perturbation strength $\mu$ is set such that $\mu \in (0, \frac{\alpha}{\max_{x \in \mathcal{X}} \|x\|})$, where $\alpha$ is the constant in Theorem 3.1. Then, the minimax strategy $x^\mu$ of the corresponding asymmetrically perturbed game* (3) *satisfies $x^\mu \in \mathcal{X}^*$.*

Thus, whenever $\mu$ is below this game-dependent threshold, $x^\mu$ coincides with the minimax strategy of the original game. Figure 1b illustrates this feature in a simple example (in that example, invariance holds for $\mu < 2.5$). The asymmetric perturbation is structurally similar to Nesterov's smoothing technique (Nesterov, 2005), which also adds a strongly convex term to only one side of a minimax formulation. While that technique was originally introduced for the minimization of nonsmooth convex functions, Section 4.1 of Nesterov (2005) treats bilinear saddle-point problems as an

application. More specifically, Nesterov perturbs only the opponent's payoff, whereas we perturb only the payoff of the player being optimized. That said, the purpose of the perturbation differs in our setting, where we exploit this asymmetry to obtain the equilibrium invariance stated in Corollary 3.2.

**Remark 3.3** (Invariance without knowing the game-dependent constant)**.** The invariance result above is guaranteed when $\mu$ lies below a *game-dependent* upper bound, as characterized by $\alpha$. Nevertheless, this does not prevent us from exploiting invariance in practice for two reasons. First, we empirically observe that the invariance phenomenon appears to persist for a reasonably wide range of choices of $\mu$ in our experiments (see Figures 1b and 4). Second, in Section 4.2, we show how Corollary 3.2 can be leveraged to construct a parameter-free procedure that does not require knowledge of $\alpha$.

The key ingredient in proving Theorem 3.1 is the near-linear behavior of the objective function $g(x) := \max_{y \in \mathcal{Y}} x^\top A y$ for player $x$ in the original game. Specifically, by Claims 1–5 in Theorem 5 of Wei et al. (2021), there exists a constant $\alpha > 0$ such that:

$$\forall x^* \in \mathcal{X}^*, \; g(x) - g(x^*) \geq \alpha \cdot \operatorname{dist}(x, \mathcal{X}^*). \quad (6)$$

This inequality implies that deviating from the minimax strategy set $\mathcal{X}^*$ results in an increase in the objective function proportionally to the distance $\operatorname{dist}(x, \mathcal{X}^*)$. In contrast, the variation (i.e., the gradient) of the perturbation payoff

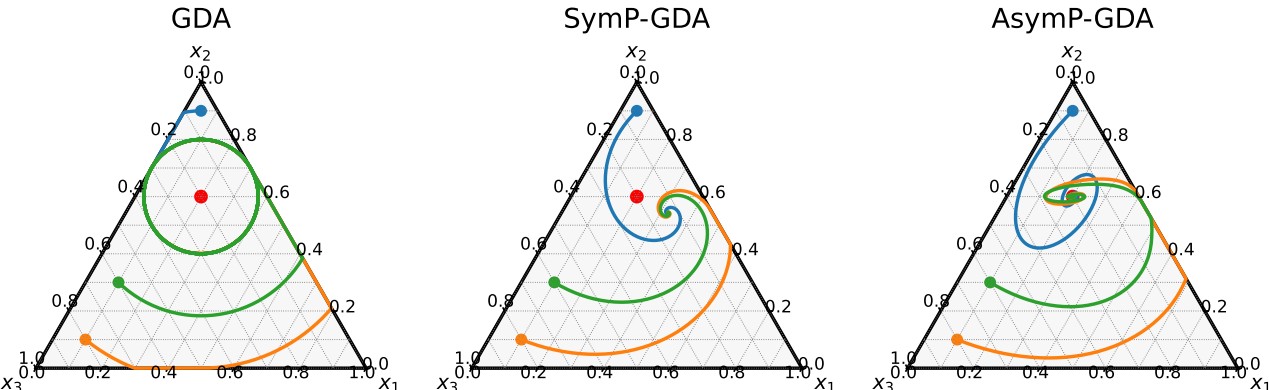

*Figure 3.* Trajectories of strategies for player $x$ using AsymP-GDA, SymP-GDA, and GDA. The learning rate is set to $\eta = 0.002$ for all methods, and the perturbation strength is set to $\mu = 1.5$ for AsymP-GDA and SymP-GDA. The game matrix $A$ is set to $A = [[0, 1, -3], [-1, 0, 1], [3, -1, 0]]$, and the strategy spaces are set to $\mathcal{X} = \mathcal{Y} = \Delta^3$. The red point represents the minimax strategy of the original game. The trajectories originate from different initial strategies, demonstrating the learning dynamics under each method.

function $\frac{\mu}{2} \|x\|^2$ can always be bounded by $\mathcal{O}(\mu)$ over $\mathcal{X}$. Hence, by choosing $\mu$ sufficiently small, we can ensure that the perturbation payoff function does not significantly incentivize player $x$ to deviate from $x^*$. In Figure 2, we illustrate this fact intuitively. The addition of the strongly convex function $\frac{\mu}{2}\|x\|^2$ does not shift the optimum $x^*$ of the original $g(x)$ if $\mu$ is sufficiently small, as the kink of the lines through $g(x)$ dominates. The detailed proof of Theorem 3.1 is provided in Appendix F.

In summary, to compute a minimax strategy $x^* \in \mathcal{X}^*$ in the original game, it is sufficient to solve the asymmetrically perturbed game (3) with a small perturbation strength $\mu > 0$. Note that, as mentioned above, one can also compute a maximin strategy $y^* \in \mathcal{Y}^*$ by solving the game $\max_{y \in \mathcal{Y}} \min_{x \in \mathcal{X}} \left\{ x^\top A y - \frac{\mu}{2} \|y\|^2 \right\}$, where the payoff perturbation is applied only to player $y$.

## 4. Asymmetrically Perturbed Gradient Descent Ascent

This section proposes a first-order method, Asymmetrically Perturbed Gradient Descent Ascent (AsymP-GDA), for solving asymmetrically perturbed games (3). At each iteration $t \in [T]$, AsymP-GDA updates each player's strategy according to the following alternating updates[4]:

$$
\begin{aligned}
x^{t+1} &= \Pi_{\mathcal{X}} \left( x^t - \eta \left( A y^t + \mu x^t \right) \right), \\
y^{t+1} &= \Pi_{\mathcal{Y}} \left( y^t + \eta A^\top x^{t+1} \right),
\end{aligned}
\tag{7}
$$

where $\eta > 0$ is the learning rate. In AsymP-GDA, player $x$'s strategy $x^t$ is updated based on the gradient of the perturbed

[4]AsymP-GDA employs alternating updates rather than simultaneous updates, as recent work has demonstrated the advantages of the former over the latter (Lee et al., 2024).

payoff function $x^\top A y + \frac{\mu}{2} \|x\|^2$, while player $y$'s strategy $y^t$ is updated using the gradient of the original payoff function $x^\top A y$. AsymP-GDA adds only negligible per-iteration runtime or memory overhead relative to standard alternating GDA, with the only additional operation being a single vector addition.

Since the perturbed payoff function of player $x$ is strongly convex, it is anticipated that AsymP-GDA enjoys a last-iterate convergence guarantee. By combining this observation with Corollary 3.2, when $\mu$ is sufficiently small, the updated strategy $x^t$ should converge to a minimax strategy $x^*$ in the original game. We confirm this empirically by plotting the trajectory of $x^t$ updated by AsymP-GDA in a sample normal-form game, as shown in Figure 3. We also provide the trajectories of GDA and SymP-GDA; in the latter, the squared $\ell^2$-norm perturbs both players' gradients. For both AsymP-GDA and SymP-GDA, the perturbation strength is set to $\mu = 1.5$. As expected, AsymP-GDA successfully converges to the minimax strategy (red point) in the original game, whereas SymP-GDA converges to a point far from the minimax strategy, and GDA cycles around the minimax strategy. Further details and additional experiments in normal-form games can be found in Appendix A.3.

### 4.1. Last-Iterate Convergence Rate

In this section, we provide a last-iterate convergence result for AsymP-GDA. Let $\|A\|$ denote the largest singular value of the matrix $A$. As discussed in the previous section, when both players' objectives are perturbed by strongly convex penalties, standard first-order methods converge to an approximate equilibrium at a linear rate (Cen et al., 2021; 2023; Pattathil et al., 2023; Sokota et al., 2023). Theorem 4.1 shows that, perhaps surprisingly, a linear convergence rate can still be achieved even when the perturbation is applied

only to one player:

**Theorem 4.1.** *For an arbitrary perturbation strength $\mu > 0$, if the learning rate satisfies $\eta \leq \frac{\mu}{\mu^2 + \|A\|^2}$, then $z^t = (x^t, y^t)$ satisfy:*

$$\mathrm{dist}(z^t, \mathcal{Z}^\mu)^2 \leq \left( \frac{1}{1 + \eta^2/(2\beta_\mu^2)} \right)^{t-1} \mathrm{dist}(z^1, \mathcal{Z}^\mu)^2,$$

*where $\mathcal{Z}^\mu := \{x^\mu\} \times \mathcal{Y}^\mu$ and $\beta_\mu > 0$ is a constant independent of $\eta$, defined explicitly in (24) in Appendix G.1.*

Note that Theorem 4.1 holds for any fixed $\mu > 0$. Hence, combining Corollary 3.2 with Theorem 4.1, we can conclude that if $\mu$ is sufficiently small (which does not need to depend on the number of iterations $t$ or $T$), player $x$'s strategy $x^t$ updated by AsymP-GDA converges to a minimax strategy of the original game (1):

**Corollary 4.2.** *Assume that the perturbation strength $\mu$ is set such that $\mu \in (0, \frac{\alpha}{\max_{x \in \mathcal{X}} \|x\|})$, and the learning rate is set so that $\eta \leq \frac{\mu}{\mu^2 + \|A\|^2}$. Then, AsymP-GDA ensures the convergence of $x^t$ to an equilibrium $x^*$ in the original game at a linear rate:*

$$\left\| x^* - x^t \right\|^2 \leq \left( \frac{1}{1 + \eta^2/(2\beta_\mu^2)} \right)^{t-1} \mathrm{dist}(z^1, \mathcal{Z}^\mu)^2.$$

Corollary 4.2 provides a linear last-iterate convergence rate for AsymP-GDA. This rate is competitive with those of optimistic methods, such as OGDA and OMWU in certain settings (e.g., bilinear games) (Wei et al., 2021). Empirically, however, AsymP-GDA exhibits faster convergence in our experiments (see Figure 6 in Appendix A).

### 4.2. Parameter-free AsymP-GDA

Although Corollary 4.2 guarantees convergence to an equilibrium of the original game only when the perturbation strength $\mu$ is sufficiently small, the appropriate scale of $\mu$ may be unknown a priori. To eliminate this tuning requirement, we also introduce a parameter-free variant (Algorithm 1 in Appendix D) that solves asymmetrically perturbed games along a sequence of perturbation strengths $\mu \in (0, \mu_{\mathrm{init}}]$, starting from an arbitrarily large $\mu_{\mathrm{init}} > 0$. For each perturbed game, the algorithm runs AsymP-GDA until the duality gap falls below a prescribed threshold, and then checks whether the resulting strategy profile has a Nash-Conv value at most $\varepsilon$. If not, it halves $\mu$ and moves on to the next perturbed game.

Thanks to Corollary 3.2, it suffices to solve only finitely many perturbed games, regardless of any target accuracy $\varepsilon$. Since each perturbed game can be solved at a linear rate by AsymP-GDA, the overall procedure retains a linear rate to an equilibrium of the original game:

**Theorem 4.3.** *For any target accuracy $\varepsilon > 0$ and any initialization $\mu_{\mathrm{init}} > 0$, Algorithm 1 returns a strategy profile $(x, y)$ with $\mathrm{NashConv}(x, y) \leq \varepsilon$ after at most $\mathcal{O}(\ln(1/\varepsilon))$ iterations in total.*

**Remark 4.4** (Comparison with decreasing-$\mu$ symmetric approaches). Whereas Liu et al. (2023) adopt a similar decreasing-$\mu$ mechanism, their analysis yields an $\tilde{\mathcal{O}}(1/\varepsilon)$ iteration complexity to reach NashConv at most $\varepsilon$. This gap stems from the fact that our decreasing-$\mu$ procedure requires solving only a bounded number of perturbed games independent of the target accuracy $\varepsilon$, while in Liu et al. (2023) the number of perturbed games grows as one targets higher accuracy. That said, this bounded number depends on the game instance and can in principle be arbitrarily large, since it scales with how small the game-dependent threshold $\alpha/\max_{x \in \mathcal{X}} \|x\|$ in Corollary 3.2 is. In Appendix E, we exhibit a concrete family of games on which this threshold becomes arbitrarily small. Consequently, in the worst case, the number of iterations required by our procedure may exceed that of symmetric approaches.

### 4.3. Proof Sketch of Theorem 4.1

This section outlines the proof sketch for Theorem 4.1. The complete proofs are provided in Appendix G.

**(1) Monotonic decrease of the distance function.** Firstly, leveraging the strong convexity of the perturbation payoff function, $\frac{\mu}{2} \|x\|^2$, we can show that the distance between the current strategy profile $z^t = (x^t, y^t)$ and any equilibrium $z^\mu = (x^\mu, y^\mu)$ monotonically decreases under $\eta \leq \frac{\mu}{\mu^2 + \|A\|^2}$. Specifically, we have for any $t \geq 1$:

$$\left\| z^\mu - z^{t+1} \right\|^2 - \left\| z^\mu - z^t \right\|^2$$
$$\leq -\left( \eta\mu - \eta^2 \left( \mu^2 + \|A\|^2 \right) \right) \left\| x^\mu - x^{t+1} \right\|^2 - \frac{1}{2} \left\| z^{t+1} - z^t \right\|^2.$$
$$(8)$$

**(2) Lower bound on the path length.** The primary technical challenge is deriving the term related to the distance between $y^{t+1}$ and the maximin strategies set $\mathcal{Y}^\mu$, i.e., $\mathrm{dist}(y^{t+1}, \mathcal{Y}^\mu)^2$, which leads to the last-iterate convergence rate. To this end, we derive a lower bound on the path length $\left\| z^{t+1} - z^t \right\|$ in terms of the distance from $z^t$ to the equilibrium set $\mathcal{Z}^\mu := \{x^\mu\} \times \mathcal{Y}^\mu$ (as shown in Lemma G.1).

Let us represent the update rule of AsymP-GDA in (7) as $z^{t+1} = T_{\eta,\mu}(z^t)$ using the operator $T_{\eta,\mu} : \mathcal{Z} \to \mathcal{Z}$. The key step is to show that the tangent residual of the perturbed game (3) is lower-bounded by the distance to the equilibrium set $\mathcal{Z}^\mu$, which we obtain by extending the classical Hoffman's bound (Hoffman, 1952) (as shown in Lemma G.2). Combining this with the first-order optimality condition for the Euclidean projection in $T_{\eta,\mu}$ yields the following

error bound:

$$\forall z \in \mathcal{Z}, \ \operatorname{dist}(z, \mathcal{Z}^\mu) \leq \frac{\beta_\mu}{\eta} \left\| z - T_{\eta,\mu}(z) \right\|,$$

where $\beta_\mu > 0$ is a constant independent of $\eta$. By setting $z = z^t$ in the above inequality, we obtain the following lower bound on $\left\| z^{t+1} - z^t \right\|$ by the distance between the current strategy profile and the equilibrium set:

$$\left\| z^{t+1} - z^t \right\| \geq \frac{\eta}{\beta_\mu} \operatorname{dist}(z^t, \mathcal{Z}^\mu). \tag{9}$$

**(3) Last-iterate convergence rate.** Putting (9) into (8), we have for any $t \geq 1$:

$$\left( 1 + \frac{\eta^2}{2\beta_\mu^2} \right) \operatorname{dist}(z^{t+1}, \mathcal{Z}^\mu)^2 \leq \operatorname{dist}(z^t, \mathcal{Z}^\mu)^2.$$

Therefore, by mathematical induction, we finally obtain the following upper bound on the distance $\operatorname{dist}(z^t, \mathcal{Z}^\mu)^2$:

$$\operatorname{dist}(z^t, \mathcal{Z}^\mu)^2 \leq \left( \frac{1}{1 + \eta^2/(2\beta_\mu^2)} \right)^{t-1} \operatorname{dist}(z^1, \mathcal{Z}^\mu)^2.$$

$\square$

**Remark 4.5** (Technical challenge in proving Theorem 4.1)**.** The main technical challenge in proving Theorem 4.1 arises from the asymmetric nature of the perturbation, which is applied only to player $x$. Unlike the symmetric case, where strong convexity in both players' payoff functions directly yields contraction, the asymmetric setting requires a more subtle analysis since player $y$'s payoff function remains linear. Our proof establishes a linear last-iterate convergence rate by lower-bounding the variation of $z^t$ by the distance from $z^t$ to the equilibrium set $\mathcal{Z}^\mu$, as shown in (9), rather than relying on strong convexity. A further complication is that, due to the projections onto $\mathcal{X}$ and $\mathcal{Y}$, the update mapping $T_{\eta,\mu}$ is not affine in general. As a result, the dynamics cannot be analyzed as a single linear system, in contrast to unconstrained linear-quadratic games (Zhang et al., 2022). See Appendix G for details.

# 5. Dilated AsymP-GDA for Extensive-Form Games

We now extend our approach to extensive-form games, which model sequential decision-making under imperfect information. To cast these games as saddle-point problems, we adopt the sequence-form representation (von Stengel, 1996). In the sequence-form representation, each player's strategy is parameterized by realization (reach) probabilities over feasible action sequences, rather than by enumerating pure strategies (i.e., complete behavior plans). Moreover, to reduce the per-iteration computational cost of AsymP-GDA in the sequence-form representation, we use the dilated Euclidean regularizer (Hoda et al., 2010) in the following experiments.

**Sequence-form representation.** We consider two-player zero-sum extensive-form games with perfect recall. Let $\mathcal{I}_x$ and $\mathcal{I}_y$ denote the sets of information sets of players $x$ and $y$, respectively. For each information set $i \in \mathcal{I}_x$, let $A(i)$ be the set of actions available at $i$.

We use the sequence-form representation to express each player's strategy. For player $x$, we write:

$$x = (x_{i,a})_{i \in \mathcal{I}_x, a \in A(i)},$$

where $x_{i,a} \geq 0$ denotes the reach probability of taking action $a$ at information set $i$, counting only the contribution of player $x$'s own action probabilities. To express the constraints for the strategy space $\mathcal{X}$, for each information set $i \in \mathcal{I}_x$, we define $\operatorname{parent}(i)$ to be the unique parent (predecessor) pair[5] $(j, b)$ (an information set $j$ and an action $b$) on player $x$'s own decision sequence that immediately precedes $i$. We also write $\operatorname{parent}(i) = \varnothing$ if no such predecessor exists, namely if $i$ is a root information set for player $x$ along its own decision sequence. The strategy space $\mathcal{X}$ is then given by the following constraints:

$$\forall i \in \mathcal{I}_x, \ \sum_{a \in A(i)} x_{i,a} = x_{\operatorname{parent}(i)},$$

$$\forall i \in \mathcal{I}_x, \ \forall a \in A(i), \ x_{i,a} \geq 0,$$

where $x_{\operatorname{parent}(i)}$ denotes $x_{j,b}$ if $\operatorname{parent}(i) = (j, b)$, and denotes 1 if $\operatorname{parent}(i) = \varnothing$. We define $\mathcal{Y}$ analogously for player $y$.

In the sequence-form representation, the two-player zero-sum extensive-form game admits a bilinear saddle-point formulation, i.e., there exists a matrix $A$ such that the game can be written as $\min_{x \in \mathcal{X}} \max_{y \in \mathcal{Y}} x^\top A y$.

**Dilated AsymP-GDA.** Due to the computational efficiency of the strategy update, we propose a variant of AsymP-GDA that employs the dilated squared Euclidean regularizer (instead of the standard squared Euclidean regularizer) for both the proximal regularizer and the perturbation term. We refer to this variant as AsymP-DGDA. For comparison, we also evaluate a symmetrically perturbed counterpart, SymP-DGDA.

For player $x$, we define the dilated squared Euclidean regularizer (Hoda et al., 2010):

$$\psi_{\operatorname{dil}}(x) = \frac{1}{2} \sum_{i \in \mathcal{I}_x} \alpha_i \sum_{a \in A(i)} \frac{x_{i,a}^2}{x_{\operatorname{parent}(i)}},$$

where $\alpha_i > 0$ is a weight parameter, and similarly for player $y$. Using $\psi_{\operatorname{dil}}$, AsymP-DGDA updates each player's strategy

---

[5]The uniqueness of the predecessor is guaranteed by the perfect recall assumption.

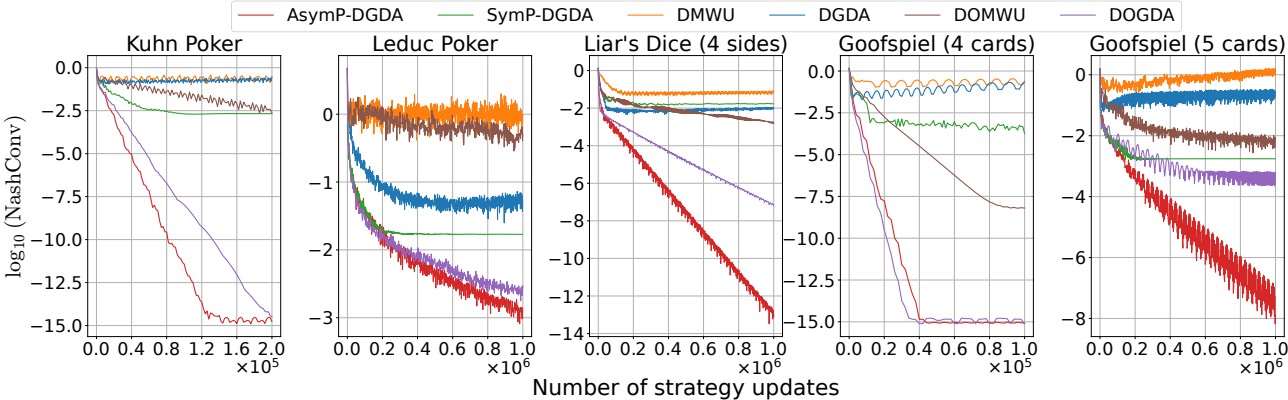

*Figure 4.* Performance in extensive-form games. The y-axis reports the NashConv of the strategy pair $(x^t, y^t)$ in the original game; for AsymP-DGDA, $(x^t, y^t)$ is obtained by combining two separate runs for players $x$ and $y$. The x-axis reports the total number of strategy updates; for AsymP-DGDA, this is the sum across the two runs.

as:

$$x^{t+1} = \arg\min_{x \in \mathcal{X}} \left\{ \eta \langle x, Ay^t + \mu \nabla \psi_{\mathrm{dil}}(x^t) \rangle + D_{\psi_{\mathrm{dil}}}(x, x^t) \right\},$$

$$y^{t+1} = \arg\min_{y \in \mathcal{Y}} \left\{ \eta \langle y, -A^\top x^{t+1} \rangle + D_{\psi_{\mathrm{dil}}}(y, y^t) \right\},$$

where $D_{\psi_{\mathrm{dil}}}(a, b) = \psi_{\mathrm{dil}}(a) - \psi_{\mathrm{dil}}(b) - \langle \nabla \psi_{\mathrm{dil}}(b), a - b \rangle$. In contrast, SymP-DGDA applies perturbation symmetrically to both players. As with AsymP-GDA, AsymP-DGDA adds only negligible per-iteration runtime or memory overhead relative to standard Dilated GDA.

**Remark 5.1** (Technical challenge in extending Theorem 4.1 to AsymP-DGDA)**.** Our proof of Theorem 4.1 relies on the global smoothness of the standard squared Euclidean regularizer over the strategy space. In contrast, the smoothness constant of the dilated squared Euclidean regularizer $\psi_{\mathrm{dil}}$ depends on inverse parent reach probabilities and may become arbitrarily large near the boundary of $\mathcal{X}$. Hence, an analogous global bound is generally unavailable, as also noted by Lee et al. (2021). We leave showing convergence for AsymP-DGDA as a promising direction.

**Empirical performance.** We compare the NashConv in the original game of the last-iterate strategy for AsymP-DGDA against SymP-DGDA and baseline algorithms, including Dilated MWU (DMWU), Dilated GDA (DGDA)[6], Dilated OMWU (Kroer et al., 2020; Lee et al., 2021), and Dilated OGDA (Farina et al., 2019; Lee et al., 2021). Our experiments focus on five different extensive-form games: Kuhn Poker, Leduc Poker, Liar's Dice (with four-sided), and Goofspiel (with four-card and five-card variants), all of which are implemented using LiteEFG (Liu et al., 2024).

---

[6]DMWU corresponds to mirror descent on the sequence-form strategy space with the dilated entropy regularizer $\psi_{\mathrm{dil\text{-}ent}}(x) = \sum_{i \in \mathcal{I}_x} \alpha_i \sum_{a \in A(i)} x_{i,a} \ln \frac{x_{i,a}}{x_{\mathrm{parent}(i)}}$. DGDA corresponds to mirror descent with the dilated Euclidean regularizer.

The detailed hyperparameters of the algorithms, tuned for best performance, are shown in Table 4 in Appendix A.

To recover equilibrium strategies for both players in the original game, we run AsymP-DGDA separately for each player: One run outputs player $x$'s strategy sequence $\{x^t\}_{t=1}^T$, and another run (with the player roles flipped) outputs player $y$'s strategy sequence $\{y^t\}_{t=1}^T$. At each iteration $t$, we then form the strategy pair $(x^t, y^t)$ by combining these outputs, and the plotted quantity is the NashConv of this pair in the original game. For a fair comparison, we measure the cost of AsymP-DGDA as the total number of strategy updates aggregated across the two runs. Equivalently, each run is given only half of the total iteration budget, so that the overall strategy-update cost matches that of the other methods.

Figure 4 shows the NashConv values for each game. As indicated by these results, AsymP-DGDA not only converges with a competitive or faster speed than any other method in all games but also directly reaches an equilibrium strategy, whereas SymP-DGDA converges near the equilibrium. These results confirm that the asymmetric perturbation leads to convergence in extensive-form games.

## 6. Related Literature

Saddle-point optimization problems have attracted significant attention due to their applications in machine learning, such as training generative adversarial networks (Daskalakis et al., 2018). No-regret learning algorithms have been extensively studied with the aim of achieving either average-iterate or last-iterate convergence. To attain last-iterate convergence, many recent algorithms incorporate optimism (Rakhlin & Sridharan, 2013a;b), including optimistic multiplicative weights update (Daskalakis & Panageas, 2019; Lei et al., 2021; Wei et al., 2021), optimistic gradient

descent ascent (Daskalakis et al., 2018; Mertikopoulos et al., 2019; de Montbrun & Renault, 2022), and extra-gradient methods (Golowich et al., 2020; Mokhtari et al., 2020).

As an alternative approach, payoff perturbation has gained renewed attention. In this approach, players' payoff functions are regularized with strongly convex terms (Cen et al., 2021; 2023; Pattathil et al., 2023), which stabilizes the dynamics and leads to convergence. Some existing works have shown convergence to an approximate equilibrium under fixed perturbation (Sokota et al., 2023; Tuyls et al., 2006; Coucheney et al., 2015; Leslie & Collins, 2005; Abe et al., 2022; Hussain et al., 2023). To recover equilibria of the original game, later studies have employed a decreasing schedule or iterative regularization (Facchinei & Pang, 2003; Koshal et al., 2013; Yousefian et al., 2017; Bernasconi et al., 2024; Liu et al., 2023; Cai et al., 2023), or have updated the regularization center periodically (Perolat et al., 2021; Abe et al., 2023; 2024). In contrast to these approaches, our algorithms avoid iteration-budget-dependent tuning of the perturbation strength by exploiting the equilibrium invariance property of our asymmetric perturbation.

Extensive-form games, which model sequential decision-making under imperfect information, have been studied extensively from both theoretical and empirical perspectives. Regarding last-iterate convergence, Lee et al. (2021) establish last-iterate convergence guarantees for optimistic algorithms in the sequence-form representation (von Stengel, 1996). More recently, Liu et al. (2023) show last-iterate convergence for symmetrically perturbed algorithms with a decreasing schedule on the perturbation strength. In contrast, our asymmetric perturbation avoids such delicate scheduling of the perturbation strength, and thus may be more amenable to large-scale extensive-form settings where optimistic algorithms can be difficult to deploy reliably.

## 7. Conclusion and Limitations

This paper introduces an asymmetric perturbation technique for solving saddle-point optimization problems, addressing key challenges in learning dynamics and equilibrium computation. Unlike symmetric perturbation, which in general yields only approximate equilibria for any fixed perturbation strength, our approach admits an invariance regime: for sufficiently small $\mu$, the minimax strategy of the perturbed game coincides with that of the original game. Building on this structural property, we propose AsymP-GDA and establish a linear last-iterate convergence rate, improving over symmetric perturbation approaches in the same bilinear setting. Nevertheless, AsymP-GDA still requires choosing $\mu$ within an allowable range. To overcome this, we further provide a parameter-free procedure that leverages the same invariance phenomenon to retain a linear rate without requiring knowledge of game-dependent constants.

Our theoretical results target bilinear two-player zero-sum games. The key insight of equilibrium invariance in Corollary 3.2 relies on the near-linear growth of the objective function. We believe that an analogous formulation can be posed beyond bilinear games, including two-player zero-sum Markov games.

Corollary 3.2 only guarantees equilibrium invariance for one player. Hence, recovering a strategy pair for both players in the original game requires running AsymP-GDA twice, once for each player. Moreover, the allowable range of $\mu$ in this result can be arbitrarily small in the worst case, although our parameter-free procedure does not require knowing this range in advance. Addressing these limitations is an important direction for future work.

## Acknowledgements

Kaito Ariu is supported by JSPS KAKENHI Grant Number JP25K21291.

## Impact Statement

This paper presents work whose goal is to solve saddle-point optimization problems. There are many potential societal consequences of our work, none of which we feel must be specifically highlighted here.

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

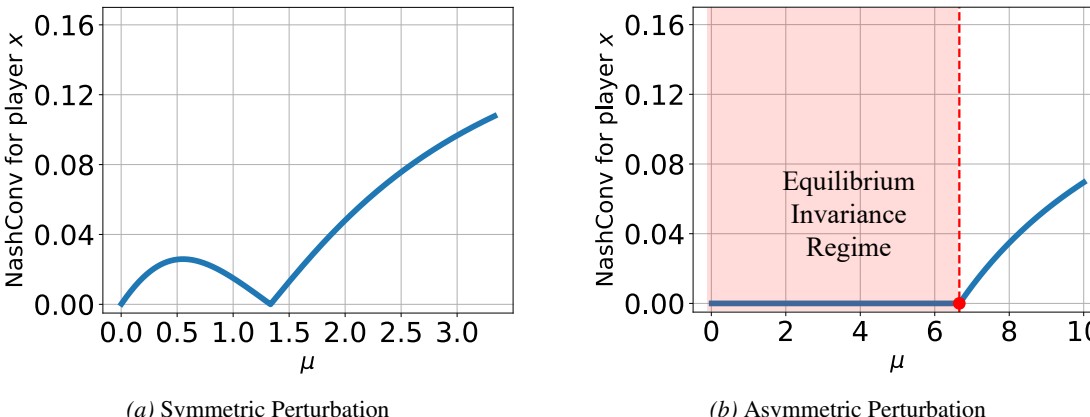

*(a)* Symmetric Perturbation    *(b)* Asymmetric Perturbation

*Figure 5.* The proximity of $x^\mu$ to $x^*$ under the symmetric perturbation and the asymmetric perturbation with varying $\mu$. The game matrix $A$ is given by $\left[\left[\frac{1}{3}, -\frac{2}{3}\right], \left[-\frac{2}{3}, 1\right]\right]$.

## A. Experimental Details and Additional Experimental Results

### A.1. Information on the Computer Resources

All experiments in this paper were conducted on macOS Sonoma 14.4.1 with Apple M2 Max and 32GB RAM.

### A.2. Proximity to Equilibrium under Symmetric and Asymmetric Perturbations in Biased Matching Pennies

This section investigates the proximity of $x^\mu$ to the equilibrium $x^*$ in the Biased Matching Pennies (BMP) game under the symmetric/asymmetric payoff perturbation, with varying perturbation strength $\mu$. The game matrix for BMP is provided in Table 1.

*Table 1.* Game matrix in BMP

|       | $y_1$  | $y_2$  |
|-------|--------|--------|
| $x_1$ | 1/3    | −2/3   |
| $x_2$ | −2/3   | 1      |

BMP has a unique equilibrium $x^* = y^* = \left(\frac{5}{8}, \frac{3}{8}\right)$, and the game value is given as $v^* = -\frac{1}{24}$.

Figure 5 exhibits the proximity of $x^\mu$ to $x^*$ as $\mu$ varies. Notably, under the symmetric perturbation, $x^\mu$ coincides with $x^*$ when $\mu$ is set to $\mu = \frac{\left(\frac{1_m}{m}\right)^\top A\left(\frac{1_n}{n}\right) - v^*}{\|x^*\|^2 - \frac{1}{m}} = \frac{4}{3}$. This result underscores the statement in Theorem B.1, that $x^\mu$ does not coincide with $x^*$ as long as $\mu \neq \frac{\left(\frac{1_m}{m}\right)^\top A\left(\frac{1_n}{n}\right) - v^*}{\|x^*\|^2 - \frac{1}{m}}$.

### A.3. Additional Experiments in Normal-Form Games

In this section, we experimentally compare our AsymP-GDA with SymP-GDA, GDA, and OGDA (Daskalakis et al., 2018; Wei et al., 2021). We conduct experiments on two normal-form games: Biased Rock-Paper-Scissors (BRPS) and Multiple Nash Equilibria (M-Ne). These games are taken from Abe et al. (2023) and Wei et al. (2021). Tables 2 and 3 provide the game matrices for BRPS and M-Ne, respectively.

Figure 6 illustrates the logarithm of NashConv averaged over 100 different random seeds. For each random seed, the initial strategies $(x^0, y^0)$ are chosen uniformly at random within the strategy spaces $\mathcal{X} = \Delta^m$ and $\mathcal{Y} = \Delta^n$. We use a learning rate of $\eta = 0.01$ for each algorithm, and a perturbation strength of $\mu = 1$ for both AsymP-GDA and SymP-GDA. We observe that AsymP-GDA converges to the minimax strategy of the original game, while SymP-GDA converges to a point far from the minimax strategy.

Figure 7 illustrates the trajectories of SymP-GDA (top row) and AsymP-GDA (bottom row) under varying perturbation

*Table 2.* Game matrix in BRPS

|       | $y_1$ | $y_2$ | $y_3$ |
|-------|-------|-------|-------|
| $x_1$ | 0     | 1     | $-3$  |
| $x_2$ | $-1$  | 0     | 1     |
| $x_3$ | 3     | $-1$  | 0     |

*Table 3.* Game matrix in M-Ne

|       | $y_1$ | $y_2$ | $y_3$ | $y_4$ | $y_5$ |
|-------|-------|-------|-------|-------|-------|
| $x_1$ | 0     | $-1$  | 1     | 0     | 0     |
| $x_2$ | 1     | 0     | $-1$  | 0     | 0     |
| $x_3$ | $-1$  | 1     | 0     | 0     | 0     |
| $x_4$ | $-1$  | 1     | 0     | 2     | $-1$  |
| $x_5$ | $-1$  | 1     | 0     | $-1$  | 2     |

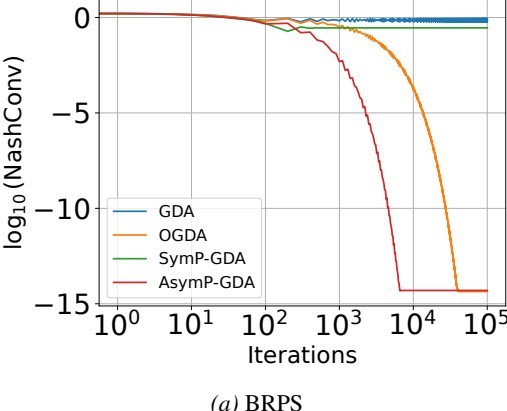

*(a)* BRPS

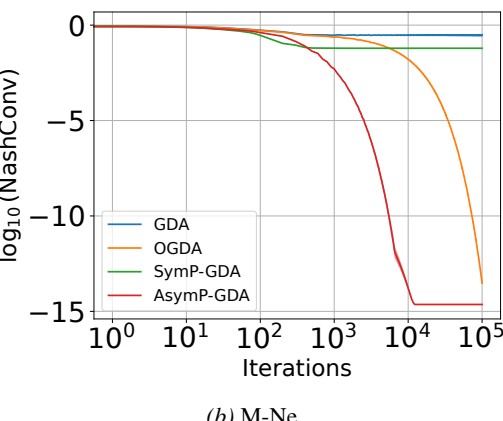

*(b)* M-Ne

*Figure 6.* Performance of AsymP-GDA, SymP-GDA, GDA, and OGDA in normal-form games. The shaded area represents the standard errors.

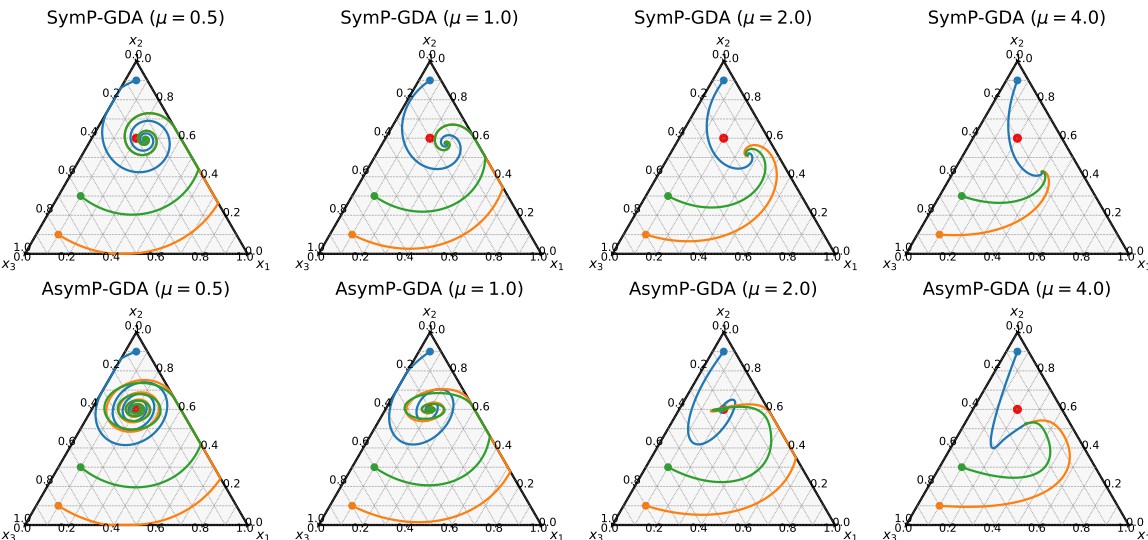

*Figure 7.* Trajectories for SymP-GDA (top row) and AsymP-GDA (bottom row) under different perturbation strengths $\mu \in \{0.5, 1.0, 2.0, 4.0\}$ in BRPS. The learning rate is set to $\eta = 0.002$ for both methods.

strengths $\mu \in \{0.5, 1.0, 2.0, 4.0\}$ in BRPS. For SymP-GDA, the trajectories do not converge directly to the equilibrium even for small values of $\mu = 0.5, 1.0$. Instead, they follow circuitous and elongated paths, resulting in slower convergence. Conversely, as $\mu$ increases ($\mu = 2.0, 4.0$), the trajectories become more direct, leading to faster convergence, but they remain farther from the equilibrium. In contrast, AsymP-GDA leads to direct convergence to the equilibrium with small perturbation strengths. For $\mu$ values up to $2.0$ the trajectories converge directly to the equilibrium. However, as $\mu$ increases beyond a threshold ($\mu = 4.0$), the trajectory deviates from the equilibrium. These results provide a more detailed understanding of the trends observed in Figures 1a and 1b, further illustrating the differences in convergence dynamics between symmetric and

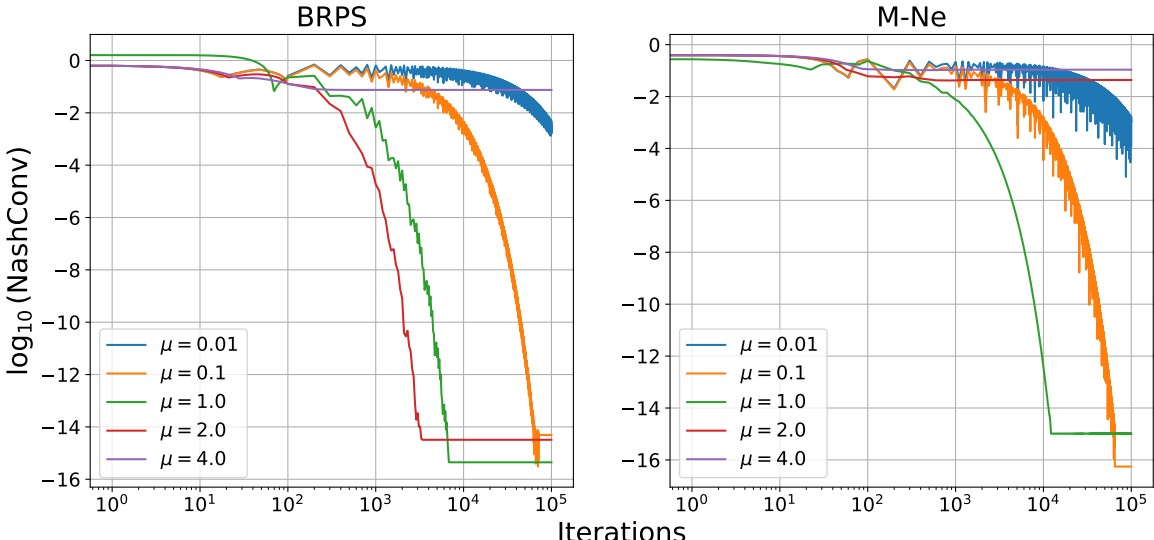

*Figure 8.* Sensitivity to the perturbation strength for AsymP-GDA with $\eta = 0.01$ in BRPS and M-Ne.

asymmetric perturbations.

Figure 8 illustrates the performance of AsymP-GDA on BRPS and M-Ne with $\mu \in \{0.01, 0.1, 1.0, 2.0, 4.0\}$ and $\eta = 0.01$. For sufficiently small $\mu$, the limit point coincides with an equilibrium of the original game. However, decreasing $\mu$ also slows convergence. Overall, these results highlight a trade-off between accuracy and convergence speed.

### A.4. Hyperparameter Settings for Extensive-Form Games

For the experiments in Section 5, we tuned the hyperparameters of each algorithm separately for each game to obtain the best performance. The resulting settings are summarized in Table 4. For the dilated regularizer, we use the unweighted version throughout, setting $\alpha_i = 1$ for all $i$, as in Lee et al. (2021).

### A.5. Comparison with CFR-Based Algorithms

This section compares the performance of AsymP-DGDA with several CFR-based algorithms, including CFR (Zinkevich et al., 2007), CFR+ (Tammelin, 2014), Discounted CFR (DCFR), and Linear CFR (LCFR) (Brown & Sandholm, 2019). We use the same games as in Section 5 and keep the hyperparameter settings for AsymP-DGDA and SymP-DGDA unchanged. Figure 9 shows the logarithm of NashConv as a function of the number of strategy updates. For AsymP-DGDA and SymP-DGDA, we report NashConv for the last-iterate strategies. For the CFR-based algorithms, we report NashConv for the average-iterate strategies. Overall, AsymP-DGDA achieves lower NashConv than the CFR-based baselines on most games, with Leduc Poker being the exception.

## B. Impossibility Results for Symmetric Perturbation

As we stated in Section 2, existing works (Liu et al., 2023; Abe et al., 2024) have shown that the distance between the solution of the symmetrically perturbed game $(x^\mu, y^\mu)$ and the solution in the original game $(x^*, y^*)$ is upper bounded by $\mathcal{O}(\mu)$. However, they do not guarantee that the two solutions coincide, even for a small $\mu > 0$. In contrast, the following theorem provides the first formal impossibility result, proving that $(x^\mu, y^\mu)$ almost never coincides with $(x^*, y^*)$.

**Theorem B.1.** *Consider a normal-form game with a unique interior equilibrium. Assume that at this equilibrium, neither player chooses their actions uniformly at random, i.e., $x^* \neq \frac{1}{m}\mathbf{1}_m$ and $y^* \neq \frac{1}{n}\mathbf{1}_n$. Then, for any $\mu > 0$ such that $\mu \neq \frac{\left(\frac{\mathbf{1}_m}{m}\right)^\top A\left(\frac{\mathbf{1}_n}{n}\right) - v^*}{\|x^*\|^2 - \frac{1}{m}}$, the minimax strategy $x^\mu$ in (2) satisfies $x^\mu \neq x^*$. Furthermore, for any $\mu > 0$ such that $\mu \neq \frac{v^* - \left(\frac{\mathbf{1}_m}{m}\right)^\top A\left(\frac{\mathbf{1}_n}{n}\right)}{\|y^*\|^2 - \frac{1}{n}}$, the maximin strategy $y^\mu$ in (2) satisfies $y^\mu \neq y^*$.*

| Game | Algorithm | $\eta$ | $\mu$ |
|---|---|---|---|
| | DMWU | 0.01 | - |
| | DGDA | 0.01 | - |
| Kuhn Poker | DOMWU | 0.1 | - |
| | DOGDA | 0.1 | - |
| | SymP-DGDA | 0.05 | 0.001 |
| | AsymP-DGDA | 0.1 | 0.01 |
| | DMWU | 0.1 | - |
| | DGDA | 0.01 | - |
| Leduc Poker | DOMWU | 0.1 | - |
| | DOGDA | 0.1 | - |
| | SymP-DGDA | 0.05 | 0.0001 |
| | AsymP-DGDA | 0.1 | 0.0001 |
| | DMWU | 0.01 | - |
| | DGDA | 0.01 | - |
| Liars Dice (4 sides) | DOMWU | 0.1 | - |
| | DOGDA | 0.1 | - |
| | SymP-DGDA | 0.01 | 0.0001 |
| | AsymP-DGDA | 0.1 | 0.001 |
| | DMWU | 0.01 | - |
| | DGDA | 0.01 | - |
| Goofspiel (4 cards) | DOMWU | 0.1 | - |
| | DOGDA | 0.1 | - |
| | SymP-DGDA | 0.1 | 0.0001 |
| | AsymP-DGDA | 0.1 | 0.05 |
| | DMWU | 0.01 | - |
| | DGDA | 0.01 | - |
| Goofspiel (5 cards) | DOMWU | 0.1 | - |
| | DOGDA | 0.1 | - |
| | SymP-DGDA | 0.1 | 0.0001 |
| | AsymP-DGDA | 0.1 | 0.001 |

*Table 4.* Hyperparameters

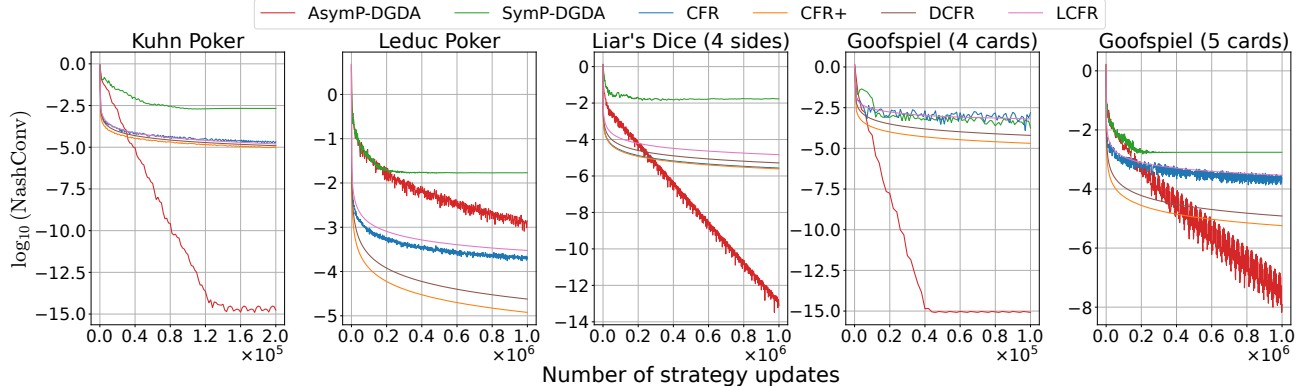

*Figure 9.* Performance in extensive-form games. AsymP-DGDA performs two strategy updates per iteration for each player. For a fair comparison across methods with different per-iteration computational costs, we report the total number of strategy updates on the x-axis rather than iterations.

The proof is provided in Appendix I. Additionally, we extend our analysis to the case where both players have different perturbation strengths, i.e., $\mu_x > 0$ and $\mu_y > 0$, as shown in Appendix C.

**Discussion on Theorem B.1.** The term $\frac{\left(\frac{\mathbf{1}_m}{m}\right)^\top A\left(\frac{\mathbf{1}_n}{n}\right) - v^*}{\|x^*\|^2 - \frac{1}{m}}$ can be interpreted as a measure of the difference between the equilibrium $(x^*, y^*)$ and the uniform random strategy profile $\left(\frac{1}{m}\mathbf{1}_m, \frac{1}{n}\mathbf{1}_n\right)$. Specifically, the numerator $\left(\frac{\mathbf{1}_m}{m}\right)^\top A\left(\frac{\mathbf{1}_n}{n}\right) - v^*$ represents the difference in the payoffs, while the denominator $\|x^*\|^2 - \frac{1}{m}$ represents the difference in the squared $\ell^2$-norms, respectively. A promising direction for future research is to theoretically demonstrate that, when $\mu = \frac{v^* - \left(\frac{\mathbf{1}_m}{m}\right)^\top A\left(\frac{\mathbf{1}_n}{n}\right)}{\|y^*\|^2 - \frac{1}{m}}$, the corresponding equilibrium coincides exactly with the equilibrium in the original game, i.e., $(x^\mu, y^\mu) = (x^*, y^*)$. We have experimentally confirmed this, and the results are presented in the Appendix A.2.

**When the game is symmetric.** Next, let us consider the case when $A^\top = -A$, as in Rock-Paper-Scissors. In this scenario, the equilibrium strategies $x^\mu$ and $y^\mu$ are not identical to the minimax or maximin strategies of the original game, regardless of the choice of $\mu > 0$.

**Corollary B.2.** *Assume that $A^\top = -A$. Under the same setup as Theorem B.1, the equilibrium $(x^\mu, y^\mu)$ in (2) always satisfies $x^\mu \neq x^*$ and $y^\mu \neq y^*$ for any $\mu > 0$.*

This is because it always holds that $v^* - \left(\frac{\mathbf{1}_m}{m}\right)^\top A\left(\frac{\mathbf{1}_n}{n}\right) = 0$ when $A^\top = -A$. Figure 1a shows the proximity of $x^\mu$ to $x^*$ with varying perturbation strength $\mu$ in a simple biased Rock-Paper-Scissors game. We observe that as long as $\mu > 0$, $x^\mu$ remains distant from $x^*$. This observation supports the theoretical results in Theorem B.1 and Corollary B.2.

## C. Independently Perturbed Game

Let us consider the perturbed game where players $x$ and $y$ choose independently their perturbation strengths $\mu_x$ and $\mu_y$:

$$\min_{x \in \mathcal{X}} \max_{y \in \mathcal{Y}} \left\{ x^\top A y + \frac{\mu_x}{2} \|x\|^2 - \frac{\mu_y}{2} \|y\|^2 \right\}. \tag{10}$$

We establish a theoretical result that contrasts with Corollary 3.2 for this perturbed game.

**Theorem C.1.** *Assume that the original game is a normal-form game with a unique interior equilibrium, and that $x^* \neq \frac{\mathbf{1}_m}{m}$ and $y^* \neq \frac{\mathbf{1}_n}{n}$, i.e., neither player chooses their actions uniformly at random. Then, for any $\mu_x > 0$ such that $\mu_x \neq \frac{\left(\frac{\mathbf{1}_m}{m}\right)^\top A\left(\frac{\mathbf{1}_n}{n}\right) - v^*}{\|x^*\|^2 - \frac{1}{m}}$, the minimax strategy $x^\mu$ in the corresponding independently perturbed game (10) satisfies $x^\mu \neq x^*$. Furthermore, for any $\mu_y > 0$ such that $\mu_y \neq \frac{v^* - \left(\frac{\mathbf{1}_m}{m}\right)^\top A\left(\frac{\mathbf{1}_n}{n}\right)}{\|y^*\|^2 - \frac{1}{n}}$, the maximin strategy $y^\mu$ in (10) satisfies $y^\mu \neq y^*$.*

## D. Parameter-Free AsymP-GDA

As noted in Remark 3.3, AsymP-GDA may require a very small perturbation strength $\mu$ to guarantee convergence to an equilibrium of the original game. To address this limitation, this section introduces a parameter-free variant (Algorithm 1) that adaptively shrinks $\mu$.

Algorithm 1 takes a target NashConv value $\varepsilon$ as an input, and runs AsymP-GDA over multiple episodes. In each episode, the algorithm runs AsymP-GDA until the duality gap of the current strategies falls below $\Theta(\varepsilon^2)$. It then checks whether the resulting strategy profile has a NashConv value at most $\varepsilon$. If the condition is satisfied, the algorithm terminates and outputs the strategies. Otherwise, it halves the perturbation strength $\mu$ and proceeds to the next episode.

### D.1. Last-Iterate Convergence Rate

We establish the last-iterate convergence rate for Algorithm 1. Without loss of generality, we assume $\max_{x \in \mathcal{X}} \|A^\top x\|_\infty \leq 1$, $\max_{y \in \mathcal{Y}} \|Ay\|_\infty \leq 1$, and $\max_{(i,j) \in [m] \times [n]} |A_{ij}| \leq 1$ since these conditions can be satisfied by re-scaling $A$. Under this assumption, Theorem 4.3 shows that the iteration complexity for the algorithm with an arbitrarily large initial perturbation strength is $\mathcal{O}(\ln(1/\varepsilon))$.

Theorem 4.3 implies that Algorithm 1 attains a linear last-iterate convergence rate, i.e., $\mathcal{O}(\exp(-t))$, even when the perturbation strength $\mu$ is initialized arbitrarily large. While Liu et al. (2023) have adopted a similar shrinking mechanism, their analysis yields a $\tilde{\mathcal{O}}(1/t)$ rate, which is slower than the linear rate established here.

---

**Algorithm 1:** Adaptive-$\mu$ AsymP-GDA

---

1   $\eta_0 \leftarrow \eta_{\text{init}}$

2   $(\hat{x}_x^{\mu_0}, \hat{y}_x^{\mu_0}) \leftarrow (x_{\text{init}}, y_{\text{init}}),\ (\hat{x}_y^{\mu_0}, \hat{y}_y^{\mu_0}) \leftarrow (x_{\text{init}}, y_{\text{init}})$

3   **for** $k = 1, 2, \cdots$ **do**

4      $\mu_k \leftarrow \frac{\mu_{\text{init}}}{2^{k-1}}$

5      $\eta_k \leftarrow \min\left(\eta_{k-1}, \frac{\mu_k}{\mu_k^2 + mn}\right)$

6      $\delta_k \leftarrow \frac{\mu_k}{2\|A\|^2 \max_{z \in \mathcal{Z}} \|z\|^2} \varepsilon^2$

7      $(x_x^1, y_x^1) \leftarrow (\hat{x}_x^{\mu_{k-1}}, \hat{y}_x^{\mu_{k-1}})$

8      $(x_y^1, y_y^1) \leftarrow (\hat{x}_y^{\mu_{k-1}}, \hat{y}_y^{\mu_{k-1}})$

9      $(\hat{x}_x^{\mu_k}, \hat{y}_x^{\mu_k}) \leftarrow \text{ASYMP-GDA}_x(x_x^1, y_x^1, \eta_k, \mu_k, \delta_k)$

10     $(\hat{x}_y^{\mu_k}, \hat{y}_y^{\mu_k}) \leftarrow \text{ASYMP-GDA}_y(x_y^1, y_y^1, \eta_k, \mu_k, \delta_k)$

11     **if** $\max_{x \in \mathcal{X}} \langle A\hat{y}_y^{\mu_k}, \hat{x}_x^{\mu_k} - x \rangle + \max_{y \in \mathcal{Y}} \langle -A^\top \hat{x}_x^{\mu_k}, \hat{y}_y^{\mu_k} - y \rangle \leq \varepsilon$ **then**

12       **return** $\hat{x}_x^{\mu_k}, \hat{y}_y^{\mu_k}$

13     **end if**

14 **end for**

---

15 **subroutine** $\text{ASYMP-GDA}_x(x^1, y^1, \eta, \mu, \delta)$

16     **for** $t = 1, 2, \cdots$ **do**

17       $x^{t+1} = \Pi_{\mathcal{X}}\left(x^t - \eta(Ay^t + \mu x^t)\right)$

18       $y^{t+1} = \Pi_{\mathcal{Y}}\left(y^t + \eta A^\top x^{t+1}\right),$

19       **if** $\max_{x \in \mathcal{X}} \langle Ay^{t+1} + \mu x^{t+1}, x^{t+1} - x \rangle + \max_{y \in \mathcal{Y}} \langle -A^\top x^{t+1}, y^{t+1} - y \rangle \leq \delta$ **then**

20         **return** $(x^{t+1}, y^{t+1})$

21     **end for**

---

22 **subroutine** $\text{ASYMP-GDA}_y(x^1, y^1, \eta, \mu, \delta)$

23     **for** $t = 1, 2, \cdots$ **do**

24       $y^{t+1} = \Pi_{\mathcal{Y}}\left(y^t + \eta(A^\top x^t - \mu y^t)\right),$

25       $x^{t+1} = \Pi_{\mathcal{X}}\left(x^t - \eta Ay^{t+1}\right)$

26       **if** $\max_{x \in \mathcal{X}} \langle Ay^{t+1}, x^{t+1} - x \rangle + \max_{y \in \mathcal{Y}} \langle -A^\top x^{t+1} + \mu y^{t+1}, y^{t+1} - y \rangle \leq \delta$ **then**

27         **return** $(x^{t+1}, y^{t+1})$

28     **end for**

---

## E. Game Instance with Arbitrarily Small Allowable Range of $\mu$

As stated in Remark 4.4, the allowable range of $\mu$ in Corollary 3.2 depends on the game instance and can in principle be arbitrarily small. We confirm this by analyzing the following normal-form game introduced by Anagnostides & Sandholm (2024), with strategy spaces $\mathcal{X} = \mathcal{Y} = \Delta^3$ and game matrix:

$$A_\gamma := \begin{pmatrix} \gamma & 0 & 0 \\ 0 & 2\gamma & 0 \\ 0 & 0 & 1 \end{pmatrix},$$

where $\gamma \in (0, 1)$. For this game, Anagnostides & Sandholm (2024) show that the unique minimax strategy is given by:

$$x^* = \left(\frac{2}{2\gamma + 3},\ \frac{1}{2\gamma + 3},\ \frac{2\gamma}{2\gamma + 3}\right)^\top.$$

The following theorem characterizes the exact range of perturbation strengths $\mu$ for which $x^\mu$ recovers $x^*$:

**Theorem E.1.** *For each $\gamma \in (0, 1)$, the minimax strategy $x^\mu$ of the asymmetrically perturbed game (3) with $A = A_\gamma$ and $\mathcal{X} = \mathcal{Y} = \Delta^3$ satisfies $x^\mu = x^*$ if and only if*

$$0 < \mu \leq \bar{\mu}(\gamma) := \frac{2\gamma(2\gamma + 3)}{1 + 4\gamma - 4\gamma^2}.$$

By Theorem E.1, the allowable range of $\mu$ can be made arbitrarily small since $\bar{\mu}(\gamma) \to 0$ as $\gamma \to 0^+$.

## F. Proofs of Theorem 3.1 and Corollary 3.2

### F.1. Proof of Theorem 3.1

*Proof of Theorem 3.1.* Let us define the function $g_{\mathrm{asym}}^{\mu} : \mathcal{X} \to \mathbb{R}$:

$$g_{\mathrm{asym}}^{\mu}(x) := \max_{y \in \mathcal{Y}} x^{\top} A y + \frac{\mu}{2} \|x\|^2 .$$

By definition, $x^{\mu} \in \arg\min_{x \in \mathcal{X}} g_{\mathrm{asym}}^{\mu}(x)$.

To bound $\mathrm{dist}(x^{\mu}, \mathcal{X}^*)$, we first introduce the following property of the function $\max_{y \in \mathcal{Y}} x^{\top} A y$:

**Lemma F.1** (Claims 1–5 in Theorem 5 of (Wei et al., 2021)). *There exists a positive constant $\alpha > 0$ such that:*

$$\forall x \in \mathcal{X}, \ \max_{y \in \mathcal{Y}} x^{\top} A y - v^* \geq \alpha \cdot \mathrm{dist}(x, \mathcal{X}^*),$$

$$\forall y \in \mathcal{Y}, \ v^* - \min_{x \in \mathcal{X}} x^{\top} A y \geq \alpha \cdot \mathrm{dist}(y, \mathcal{Y}^*),$$

*where $\alpha$ depends only on $\mathcal{X}^*$ and $\mathcal{Y}^*$.*

Since $\max_{y \in \mathcal{Y}} \left( \Pi_{\mathcal{X}^*}(x) \right)^{\top} A y = v^*$ for any $x \in \mathcal{X}$, Lemma F.1 yields:

$$\max_{y \in \mathcal{Y}} \left( \Pi_{\mathcal{X}^*}(x) \right)^{\top} A y - \max_{y \in \mathcal{Y}} x^{\top} A y \leq -\alpha \cdot \mathrm{dist}(x, \mathcal{X}^*). \tag{11}$$

We also have the following bound for any $x \in \mathcal{X}$:

$$\begin{aligned}
\frac{\mu}{2} \|\Pi_{\mathcal{X}^*}(x)\|^2 - \frac{\mu}{2} \|x\|^2 &= \frac{\mu}{2} \|\Pi_{\mathcal{X}^*}(x)\|^2 - \frac{\mu}{2} \|\Pi_{\mathcal{X}^*}(x) - (x - \Pi_{\mathcal{X}^*}(x))\|^2 \\
&= \mu \langle \Pi_{\mathcal{X}^*}(x), \Pi_{\mathcal{X}^*}(x) - x \rangle - \frac{\mu}{2} \|x - \Pi_{\mathcal{X}^*}(x)\|^2 \\
&\leq \mu \|x - \Pi_{\mathcal{X}^*}(x)\| \|\Pi_{\mathcal{X}^*}(x)\| - \frac{\mu}{2} \|x - \Pi_{\mathcal{X}^*}(x)\|^2 \\
&= \mu \, \mathrm{dist}(x, \mathcal{X}^*) \|\Pi_{\mathcal{X}^*}(x)\| - \frac{\mu}{2} \mathrm{dist}(x, \mathcal{X}^*)^2.
\end{aligned} \tag{12}$$

Adding (11) and (12), and then bounding $\|\Pi_{\mathcal{X}^*}(x)\| \leq \max_{x \in \mathcal{X}} \|x\|$, we obtain the following for any $x \in \mathcal{X}$:

$$\begin{aligned}
g_{\mathrm{asym}}^{\mu}\left( \Pi_{\mathcal{X}^*}(x) \right) - g_{\mathrm{asym}}^{\mu}(x) &\leq (\mu \|\Pi_{\mathcal{X}^*}(x)\| - \alpha) \, \mathrm{dist}(x, \mathcal{X}^*) - \frac{\mu}{2} \mathrm{dist}(x, \mathcal{X}^*)^2 \\
&\leq \left( \mu \max_{x \in \mathcal{X}} \|x\| - \alpha \right) \mathrm{dist}(x, \mathcal{X}^*) - \frac{\mu}{2} \mathrm{dist}(x, \mathcal{X}^*)^2.
\end{aligned} \tag{13}$$

Since $x^{\mu} \in \arg\min_{x \in \mathcal{X}} g_{\mathrm{asym}}^{\mu}(x)$, we have $g_{\mathrm{asym}}^{\mu}\left( \Pi_{\mathcal{X}^*}(x^{\mu}) \right) - g_{\mathrm{asym}}^{\mu}(x^{\mu}) \geq 0$. Combining this with (13) applied at $x = x^{\mu}$:

$$0 \leq g_{\mathrm{asym}}^{\mu}\left( \Pi_{\mathcal{X}^*}(x^{\mu}) \right) - g_{\mathrm{asym}}^{\mu}(x^{\mu}) \leq \mathrm{dist}(x^{\mu}, \mathcal{X}^*) \left( \mu \max_{x \in \mathcal{X}} \|x\| - \alpha - \frac{\mu}{2} \mathrm{dist}(x^{\mu}, \mathcal{X}^*) \right).$$

If $\mathrm{dist}(x^{\mu}, \mathcal{X}^*) > 0$, dividing the above inequality by $\mathrm{dist}(x^{\mu}, \mathcal{X}^*)$ yields:

$$\mathrm{dist}(x^{\mu}, \mathcal{X}^*) \leq 2 \left( \max_{x \in \mathcal{X}} \|x\| - \frac{\alpha}{\mu} \right).$$

If $\mathrm{dist}(x^{\mu}, \mathcal{X}^*) = 0$, the bound $\mathrm{dist}(x^{\mu}, \mathcal{X}^*) \leq 2 \max \left\{ 0, \max_{x \in \mathcal{X}} \|x\| - \frac{\alpha}{\mu} \right\}$ holds trivially since the right-hand side is nonnegative. Combining the two cases establishes:

$$\mathrm{dist}(x^{\mu}, \mathcal{X}^*) \leq 2 \max \left\{ 0, \max_{x \in \mathcal{X}} \|x\| - \frac{\alpha}{\mu} \right\}.$$

$\square$

### F.2. Proof of Corollary 3.2

*Proof of Corollary 3.2.* Under the assumption $\mu \in \left(0, \frac{\alpha}{\max_{x \in \mathcal{X}} \|x\|}\right)$, we have $\max_{x \in \mathcal{X}} \|x\| - \frac{\alpha}{\mu} < 0$. Hence, Theorem 3.1 implies that $\mathrm{dist}(x^\mu, \mathcal{X}^*) = 0$, which yields $x^\mu \in \mathcal{X}^*$. $\qquad\square$

## G. Proofs for Theorem 4.1

### G.1. Proof of Theorem 4.1

*Proof of Theorem 4.1.* First, we have for any vectors $a, b, c$:

$$\frac{1}{2}\|a - b\|^2 - \frac{1}{2}\|a - c\|^2 + \frac{1}{2}\|b - c\|^2 = \langle c - b, a - b \rangle. \tag{14}$$

From (14), we have for any $t \geq 1$:

$$\frac{1}{2}\left\|x^\mu - x^{t+1}\right\|^2 - \frac{1}{2}\left\|x^\mu - x^t\right\|^2 + \frac{1}{2}\left\|x^{t+1} - x^t\right\|^2 = \left\langle x^t - x^{t+1}, x^\mu - x^{t+1} \right\rangle \tag{15}$$

Here, from the first-order optimality condition for $x^{t+1}$ in (7), we have for any $t \geq 1$:

$$\left\langle \eta A y^t + \eta \mu x^t + x^{t+1} - x^t, x^{t+1} - x^\mu \right\rangle \leq 0. \tag{16}$$

Combining (15) and (16), we have for any $y^\mu \in \mathcal{Y}^\mu$:

$$\begin{aligned}
&\frac{1}{2}\left\|x^\mu - x^{t+1}\right\|^2 - \frac{1}{2}\left\|x^\mu - x^t\right\|^2 + \frac{1}{2}\left\|x^{t+1} - x^t\right\|^2 \\
&\leq \eta \left\langle A y^t + \mu x^t, x^\mu - x^{t+1} \right\rangle \\
&= \eta \left\langle A y^{t+1} + \mu x^{t+1}, x^\mu - x^{t+1} \right\rangle + \eta \left\langle A y^t - A y^{t+1} + \mu(x^t - x^{t+1}), x^\mu - x^{t+1} \right\rangle \\
&= \eta \left\langle A y^\mu + \mu x^\mu, x^\mu - x^{t+1} \right\rangle + \eta \left\langle A y^t - A y^{t+1} + \mu(x^t - x^{t+1}), x^\mu - x^{t+1} \right\rangle \\
&\quad + \eta \left\langle A y^{t+1} - A y^\mu, x^\mu - x^{t+1} \right\rangle - \eta \mu \left\|x^\mu - x^{t+1}\right\|^2.
\end{aligned} \tag{17}$$

On the other hand, from the first-order optimality condition for $x^\mu$ in (4), we get:

$$\left\langle A y^\mu + \mu x^\mu, x^\mu - x^{t+1} \right\rangle \leq 0. \tag{18}$$

By combining (17) and (18), we have for any $t \geq 1$:

$$\begin{aligned}
&\frac{1}{2}\left\|x^\mu - x^{t+1}\right\|^2 - \frac{1}{2}\left\|x^\mu - x^t\right\|^2 + \frac{1}{2}\left\|x^{t+1} - x^t\right\|^2 \\
&\leq -\eta \mu \left\|x^\mu - x^{t+1}\right\|^2 + \eta \left\langle A y^t - A y^{t+1}, x^\mu - x^{t+1} \right\rangle + \eta \mu \left\langle x^t - x^{t+1}, x^\mu - x^{t+1} \right\rangle \\
&\quad + \eta \left\langle A y^{t+1} - A y^\mu, x^\mu - x^{t+1} \right\rangle.
\end{aligned} \tag{19}$$

Similar to (15), we have for any $t \geq 1$:

$$\frac{1}{2}\left\|y^\mu - y^{t+1}\right\|^2 - \frac{1}{2}\left\|y^\mu - y^t\right\|^2 + \frac{1}{2}\left\|y^{t+1} - y^t\right\|^2 = \left\langle y^t - y^{t+1}, y^\mu - y^{t+1} \right\rangle, \tag{20}$$

and from the first-order optimality condition for $y^{t+1}$ in (7), we have for any $t \geq 1$:

$$\left\langle -\eta A^\top x^{t+1} + y^{t+1} - y^t, y^{t+1} - y^\mu \right\rangle \leq 0. \tag{21}$$

By combining (20) and (21), we get:

$$\begin{aligned}
&\frac{1}{2}\left\|y^\mu - y^{t+1}\right\|^2 - \frac{1}{2}\left\|y^\mu - y^t\right\|^2 + \frac{1}{2}\left\|y^{t+1} - y^t\right\|^2 \\
&\leq -\eta \left\langle A^\top x^{t+1}, y^\mu - y^{t+1} \right\rangle \\
&= -\eta \left\langle A^\top x^\mu, y^\mu - y^{t+1} \right\rangle - \eta \left\langle A^\top x^{t+1} - A^\top x^\mu, y^\mu - y^{t+1} \right\rangle \\
&\leq -\eta \left\langle A^\top x^{t+1} - A^\top x^\mu, y^\mu - y^{t+1} \right\rangle,
\end{aligned} \tag{22}$$

where the last inequality stems from (5).

Summing up (19) and (22), we have for any $t \geq 1$, with $z^\mu := (x^\mu, y^\mu)$:

$$\frac{1}{2} \left\| z^\mu - z^{t+1} \right\|^2 - \frac{1}{2} \left\| z^\mu - z^t \right\|^2 + \frac{1}{2} \left\| z^{t+1} - z^t \right\|^2$$

$$\leq -\eta\mu \left\| x^\mu - x^{t+1} \right\|^2 + \eta\mu \left\langle x^t - x^{t+1}, x^\mu - x^{t+1} \right\rangle + \eta \left\langle Ay^t - Ay^{t+1}, x^\mu - x^{t+1} \right\rangle$$

$$\leq -\eta\mu \left\| x^\mu - x^{t+1} \right\|^2 + \frac{1}{4} \left\| x^t - x^{t+1} \right\|^2 + \eta^2\mu^2 \left\| x^\mu - x^{t+1} \right\|^2 + \frac{1}{4} \left\| y^t - y^{t+1} \right\|^2 + \eta^2 \|A\|^2 \left\| x^\mu - x^{t+1} \right\|^2$$

$$= -\left( \eta\mu - \eta^2(\mu^2 + \|A\|^2) \right) \left\| x^\mu - x^{t+1} \right\|^2 + \frac{1}{4} \left\| z^t - z^{t+1} \right\|^2.$$

Hence, under the assumption that $\eta \leq \frac{\mu}{\mu^2 + \|A\|^2}$, we have for any $y^\mu \in \mathcal{Y}^\mu$:

$$\frac{1}{2} \left\| z^\mu - z^{t+1} \right\|^2 - \frac{1}{2} \left\| z^\mu - z^t \right\|^2 \leq -\frac{1}{4} \left\| z^t - z^{t+1} \right\|^2.$$

Defining $z_t^\mu = \arg\min_{z \in \mathcal{Z}^\mu} \|z - z^t\|$, we obtain:

$$\mathrm{dist}(z^{t+1}, \mathcal{Z}^\mu)^2 \leq \left\| z_t^\mu - z^{t+1} \right\|^2 \leq \left\| z_t^\mu - z^t \right\|^2 - \frac{1}{2} \left\| z^t - z^{t+1} \right\|^2$$

$$= \mathrm{dist}(z^t, \mathcal{Z}^\mu)^2 - \frac{1}{2} \left\| z^t - z^{t+1} \right\|^2. \tag{23}$$

Here, let us define the operator $T_{\eta,\mu} : \mathcal{Z} \to \mathcal{Z}$ as follows:

$$T_{\eta,\mu}(x, y) := \left( \Pi_\mathcal{X} \left( x - \eta \left( Ay + \mu x \right) \right), \ \Pi_\mathcal{Y} \left( y + \eta A^\top \Pi_\mathcal{X} \left( x - \eta \left( Ay + \mu x \right) \right) \right) \right).$$

Then, the update rule of AsymP-GDA in (7) can be written as $z^{t+1} = T_{\eta,\mu}(z^t)$. Regarding $T_{\eta,\mu}$, we have the following error bound:

**Lemma G.1.** *Assume that* $0 < \eta \leq \frac{\mu}{\mu^2 + \|A\|^2}$. *Then, for any* $z \in \mathcal{Z}$:

$$\mathrm{dist}(z, \mathcal{Z}^\mu) \leq \frac{\beta_\mu}{\eta} \left\| z - T_{\eta,\mu}(z) \right\|,$$

*where*

$$\beta_\mu := \left( \frac{\mu}{\mu^2 + \|A\|^2} + c_\mu \left( 1 + \frac{\mu(\mu + \|A\|)}{\mu^2 + \|A\|^2} \right) \right) \left( 1 + \frac{\mu\|A\|}{\mu^2 + \|A\|^2} \right) \tag{24}$$

*is a positive constant independent of* $\eta$, *with* $c_\mu > 0$ *being the constant obtained by applying Lemma G.2 to* $\mathcal{P} = \mathcal{Z}$, $F = F_\mu^x$, *and* $S = \mathcal{Z}^\mu$ *(depending on* $A$, $\mathcal{X}$, $\mathcal{Y}$, *and* $\mu$).

Setting $z = z^t$ in the above lemma, we obtain:

$$\|z^{t+1} - z^t\| = \|z^t - T_{\eta,\mu}(z^t)\|$$

$$\geq \frac{\eta}{\beta_\mu} \mathrm{dist}(z^t, \mathcal{Z}^\mu)$$

$$\geq \frac{\eta}{\beta_\mu} \mathrm{dist}(z^{t+1}, \mathcal{Z}^\mu),$$

where the last inequality follows from (23). Plugging the above inequality into (23), we have:

$$\left( 1 + \frac{\eta^2}{2\beta_\mu^2} \right) \mathrm{dist}(z^{t+1}, \mathcal{Z}^\mu)^2 \leq \mathrm{dist}(z^t, \mathcal{Z}^\mu)^2.$$

Therefore, when $\eta \leq \frac{\mu}{\mu^2 + \|A\|^2}$, we have for any $t \geq 1$:

$$\mathrm{dist}(z^t, \mathcal{Z}^\mu)^2 \leq \left( \frac{1}{1 + \eta^2/(2\beta_\mu^2)} \right)^{t-1} \mathrm{dist}(z^1, \mathcal{Z}^\mu)^2.$$

$\square$

### G.2. Proof of Lemma G.1

*Proof of Lemma G.1.* Throughout the proof, for a closed convex set $\mathcal{C} \subseteq \mathbb{R}^d$ and $z \in \mathcal{C}$, we let $\mathcal{N}_{\mathcal{C}}(z) := \{v \in \mathbb{R}^d \mid \langle v, z' - z \rangle \leq 0$ for all $z' \in \mathcal{C}\}$ denote the normal cone to $\mathcal{C}$ at $z$. We also define $F_{\mu}^x$ and $F_{\mu}^y$ by:

$$F_{\mu}^x(z) := \left(Ay + \mu x, \ -A^\top x\right) \quad \text{and} \quad F_{\mu}^y(z) := \left(Ay, \ -A^\top x + \mu y\right)$$

for $z = (x, y) \in \mathcal{Z}$, which correspond to the gradients of the perturbed payoff functions in (3) and its counterpart for player $y$, respectively.

Fix $z \in \mathcal{Z}$ and set $z^+ := \Pi_{\mathcal{Z}}\left(z - \eta F_{\mu}^x(z)\right)$. Since $z^+ \in \mathcal{Z}$, the triangle inequality yields:

$$\text{dist}(z, \mathcal{Z}^\mu) \leq \|z - z^+\| + \text{dist}(z^+, \mathcal{Z}^\mu). \tag{25}$$

We bound the two terms on the right-hand side separately.

To bound $\text{dist}(z^+, \mathcal{Z}^\mu)$, we introduce a Hoffman-type error bound for affine variational inequalities over a polytope:

**Lemma G.2.** *Let $\mathcal{P} \subseteq \mathbb{R}^d$ be a nonempty polytope, and let $F : \mathbb{R}^d \to \mathbb{R}^d$ be an affine mapping. Define:*

$$S := \{z \in \mathcal{P} \mid 0 \in F(z) + \mathcal{N}_{\mathcal{P}}(z)\}.$$

*Assume that $S \neq \emptyset$. Then, there exists a positive constant $H > 0$ such that for any $z \in \mathcal{P}$:*

$$\text{dist}(z, S) \leq H \min_{v \in \mathcal{N}_{\mathcal{P}}(z)} \|F(z) + v\|.$$

To apply Lemma G.2, we use the following equivalence between $\mathcal{Z}^\mu$ and the solution set of the variational inequality:

**Lemma G.3.** *For any $\mu > 0$, it holds that:*

$$\mathcal{Z}^\mu = \left\{z \in \mathcal{Z} \mid 0 \in F_{\mu}^x(z) + \mathcal{N}_{\mathcal{Z}}(z)\right\}.$$

From Lemma G.3, we can apply Lemma G.2 to $\mathcal{P} = \mathcal{Z}$, $F = F_{\mu}^x$, and $S = \mathcal{Z}^\mu$. Hence, there exists a positive constant $c_\mu > 0$, independent of $\eta$, such that:

$$\text{dist}(z^+, \mathcal{Z}^\mu) \leq c_\mu \min_{v \in \mathcal{N}_{\mathcal{Z}}(z^+)} \left\|F_{\mu}^x(z^+) + v\right\|. \tag{26}$$

By the first-order optimality condition for the Euclidean projection, we have for any $z' \in \mathcal{Z}$:

$$\left\langle z - \eta F_{\mu}^x(z) - z^+, z' - z^+ \right\rangle \leq 0.$$

Dividing by $\eta > 0$, this rewrites as:

$$\left\langle \frac{z - z^+}{\eta} - F_{\mu}^x(z), z' - z^+ \right\rangle \leq 0 \quad \text{for all } z' \in \mathcal{Z}.$$

By the definition of the normal cone, this yields:

$$\frac{z - z^+}{\eta} - F_{\mu}^x(z) \in \mathcal{N}_{\mathcal{Z}}(z^+). \tag{27}$$

Plugging $v = \frac{z - z^+}{\eta} - F_{\mu}^x(z)$ from (27) into the right-hand side of (26) yields:

$$\text{dist}(z^+, \mathcal{Z}^\mu) \leq c_\mu \left\|F_{\mu}^x(z^+) - F_{\mu}^x(z) + \frac{z - z^+}{\eta}\right\|$$

$$\leq c_\mu \left(\left\|\frac{z - z^+}{\eta}\right\| + \left\|F_{\mu}^x(z^+) - F_{\mu}^x(z)\right\|\right). \tag{28}$$

To bound the term $\left\|F_\mu^x(z^+) - F_\mu^x(z)\right\|$ on the right-hand side, we show that $F_\mu^x$ is Lipschitz continuous with constant $\mu + \|A\|$. For any $z_1 = (x_1, y_1), z_2 = (x_2, y_2) \in \mathcal{Z}$, splitting $F_\mu^x(z_1) - F_\mu^x(z_2) = \left(A(y_1 - y_2), -A^\top(x_1 - x_2)\right) + \mu(x_1 - x_2, 0)$ and applying the triangle inequality yields:

$$\left\|F_\mu^x(z_1) - F_\mu^x(z_2)\right\| \leq \left\|\left(A(y_1 - y_2), -A^\top(x_1 - x_2)\right)\right\| + \mu\|x_1 - x_2\|.$$

The first term satisfies:

$$\begin{aligned}
\left\|\left(A(y_1 - y_2), -A^\top(x_1 - x_2)\right)\right\|^2 &= \|A(y_1 - y_2)\|^2 + \|A^\top(x_1 - x_2)\|^2 \\
&\leq \|A\|^2 \left(\|y_1 - y_2\|^2 + \|x_1 - x_2\|^2\right) \\
&= \|A\|^2 \|z_1 - z_2\|^2,
\end{aligned}$$

and the second term satisfies $\mu\|x_1 - x_2\| \leq \mu\|z_1 - z_2\|$. Combining these bounds yields:

$$\left\|F_\mu^x(z_1) - F_\mu^x(z_2)\right\| \leq (\mu + \|A\|)\|z_1 - z_2\|. \tag{29}$$

Combining (28) with (29), we have:

$$\begin{aligned}
\mathrm{dist}(z^+, \mathcal{Z}^\mu) &\leq c_\mu \left(\frac{\|z - z^+\|}{\eta} + (\mu + \|A\|)\|z - z^+\|\right) \\
&= \frac{c_\mu(1 + \eta(\mu + \|A\|))\|z - z^+\|}{\eta} \\
&\leq \frac{c_\mu\left(1 + \frac{\mu(\mu + \|A\|)}{\mu^2 + \|A\|^2}\right)\|z - z^+\|}{\eta},
\end{aligned}$$

where the last inequality follows from $\eta \leq \frac{\mu}{\mu^2 + \|A\|^2}$. Plugging this into (25), we obtain:

$$\begin{aligned}
\mathrm{dist}(z, \mathcal{Z}^\mu) &\leq \|z - z^+\| + \frac{c_\mu\left(1 + \frac{\mu(\mu + \|A\|)}{\mu^2 + \|A\|^2}\right)\|z - z^+\|}{\eta} \\
&\leq \frac{\frac{\mu}{\mu^2 + \|A\|^2} + c_\mu\left(1 + \frac{\mu(\mu + \|A\|)}{\mu^2 + \|A\|^2}\right)}{\eta} \left\|z - z^+\right\|,
\end{aligned} \tag{30}$$

where the last inequality follows from $\eta \leq \frac{\mu}{\mu^2 + \|A\|^2}$.

It remains to bound $\|z - z^+\|$ by $\|z - T_{\eta,\mu}(z)\|$. Writing $z = (x, y)$, we can express $z^+$ as:

$$z^+ = \left(\Pi_{\mathcal{X}}\left(x - \eta(Ay + \mu x)\right),\ \Pi_{\mathcal{Y}}\left(y + \eta A^\top x\right)\right).$$

Hence, since $\Pi_{\mathcal{Y}}$ is nonexpansive,

$$\begin{aligned}
\left\|z^+ - T_{\eta,\mu}(z)\right\| &= \left\|\Pi_{\mathcal{Y}}\left(y + \eta A^\top x\right) - \Pi_{\mathcal{Y}}\left(y + \eta A^\top \Pi_{\mathcal{X}}(x - \eta(Ay + \mu x))\right)\right\| \\
&\leq \eta\|A\|\left\|x - \Pi_{\mathcal{X}}(x - \eta(Ay + \mu x))\right\| \\
&\leq \eta\|A\|\left\|z - T_{\eta,\mu}(z)\right\|,
\end{aligned}$$

where the last inequality uses the definition of $T_{\eta,\mu}(z)$. Therefore, by the triangle inequality,

$$\begin{aligned}
\left\|z - z^+\right\| &\leq \|z - T_{\eta,\mu}(z)\| + \left\|T_{\eta,\mu}(z) - z^+\right\| \\
&\leq (1 + \eta\|A\|)\left\|z - T_{\eta,\mu}(z)\right\| \\
&\leq \left(1 + \frac{\mu\|A\|}{\mu^2 + \|A\|^2}\right)\left\|z - T_{\eta,\mu}(z)\right\|,
\end{aligned} \tag{31}$$

where the second inequality follows from the bound on $\|z^+ - T_{\eta,\mu}(z)\|$ above, and the last inequality follows from $\eta \leq \frac{\mu}{\mu^2 + \|A\|^2}$.

Combining (30) and (31), we obtain for any $z \in \mathcal{Z}$:

$$\operatorname{dist}(z, \mathcal{Z}^\mu) \leq \frac{\beta_\mu}{\eta} \|z - T_{\eta,\mu}(z)\|,$$

with $\beta_\mu := \left( \frac{\mu}{\mu^2 + \|A\|^2} + c_\mu \left( 1 + \frac{\mu(\mu + \|A\|)}{\mu^2 + \|A\|^2} \right) \right) \left( 1 + \frac{\mu\|A\|}{\mu^2 + \|A\|^2} \right)$. $\qquad\square$

### G.3. Proof of Lemma G.2

*Proof of Lemma G.2.* Since $\mathcal{P}$ is a polytope, there exist a matrix $B \in \mathbb{R}^{p \times d}$ and a vector $b \in \mathbb{R}^p$ such that:

$$\mathcal{P} = \left\{ z \in \mathbb{R}^d \mid Bz \leq b \right\}.$$

For $z \in \mathcal{P}$, let

$$I(z) := \{ i \in [p] \mid B_i z = b_i \}$$

denote the active index set. For an index set $I \subseteq [p]$, let $B_I$ denote the submatrix of $B$ whose rows are indexed by $I$. We use the standard convention that, when $I = \emptyset$, $B_I$ is the $0 \times d$ matrix, $\mathbb{R}_+^{|I|} := \mathbb{R}_+^0 = \{0\}$, and $B_I^\top \lambda := 0$ for $\lambda \in \mathbb{R}_+^0$.

From Theorem 6.46 of Rockafellar & Wets (2009), the normal cone to $\mathcal{P}$ at $z$ is given by:

$$\mathcal{N}_\mathcal{P}(z) = \left\{ B_{I(z)}^\top \lambda \mid \lambda \in \mathbb{R}_+^{|I(z)|} \right\}. \tag{32}$$

By this representation, $z \in S$ if and only if there exists $\lambda \in \mathbb{R}_+^{|I(z)|}$ with $F(z) + B_{I(z)}^\top \lambda = 0$. To exploit this characterization while keeping the active index fixed, for each $I \subseteq [p]$ we define:

$$L_I := \left\{ (u, \gamma) \mid Bu \leq b, \ B_I u = b_I, \ \gamma \geq 0, \ F(u) + B_I^\top \gamma = 0 \right\}.$$

This is a polyhedron in the variables $(u, \gamma)$, since $F$ is affine. Let $\mathcal{P}_I := \{ z \in \mathcal{P} \mid B_I z = b_I \}$ be the face on which all constraints indexed by $I$ are tight. Taking any $(u, \gamma) \in L_I$, which by definition satisfies $B_I u = b_I$, we get $I \subseteq I(u)$. Hence, by (32),

$$B_I^\top \gamma = \sum_{i \in I} \gamma_i B_i^\top + \sum_{i \in I(u) \setminus I} 0 \cdot B_i^\top \in \mathcal{N}_\mathcal{P}(u).$$

Together with $F(u) + B_I^\top \gamma = 0$, this yields $0 = F(u) + B_I^\top \gamma \in F(u) + \mathcal{N}_\mathcal{P}(u)$, i.e., $u \in S$. This shows

$$(u, \gamma) \in L_I \Rightarrow u \in S. \tag{33}$$

From Hoffman's bound (Hoffman, 1952) for linear systems, for each $L_I$ satisfying $L_I \neq \emptyset$, there exists a positive constant $h_I > 0$ such that for any $(z, \lambda)$:

$$\min_{(u,\gamma) \in L_I} \|(z, \lambda) - (u, \gamma)\| \leq h_I \left( \|(Bz - b)_+\| + \|B_I z - b_I\| + \|(-\lambda)_+\| + \|F(z) + B_I^\top \lambda\| \right),$$

where $(a)_+$ denotes the elementwise positive part of a given vector $a$, i.e., $((a)_+)_i := \max(a_i, 0)$ for all $i$. If $z \in \mathcal{P}_I$ and $\lambda \in \mathbb{R}_+^{|I|}$, then $Bz \leq b$, $B_I z = b_I$, and $\lambda \geq 0$, so the first three violation terms vanish. Combining this with (33), we obtain for any $z \in \mathcal{P}_I$ and $\lambda \in \mathbb{R}_+^{|I|}$ satisfying $L_I \neq \emptyset$:

$$\operatorname{dist}(z, S) \leq \min_{(u,\gamma) \in L_I} \|(z, \lambda) - (u, \gamma)\| \leq h_I \|F(z) + B_I^\top \lambda\|. \tag{34}$$

On the other hand, setting $K_I := \left\{ B_I^\top \lambda \mid \lambda \in \mathbb{R}_+^{|I|} \right\}$, since the function

$$\phi_I(z) := \min_{v \in F(z) + K_I} \|v\|$$

is continuous on $\mathcal{P}_I$ and $\mathcal{P}_I$ is compact, $\phi_I(z)$ has a minimum over $\mathcal{P}_I$. Thus, if $L_I = \emptyset$ and $\mathcal{P}_I \neq \emptyset$, we can define the following positive constant:

$$\delta_I := \min_{z \in \mathcal{P}_I} \phi_I(z) > 0.$$

Hence, letting $D := \max_{z,z' \in \mathcal{P}} \|z - z'\|$ denote the diameter of $\mathcal{P}$, we obtain for $z \in \mathcal{P}_I$ and $\lambda \in \mathbb{R}_+^{|I|}$ satisfying $L_I = \emptyset$:

$$\text{dist}(z, S) \leq D = \frac{D}{\left\|F(z) + B_I^\top \lambda\right\|} \left\|F(z) + B_I^\top \lambda\right\| \leq \frac{D}{\delta_I} \left\|F(z) + B_I^\top \lambda\right\|, \tag{35}$$

where the last inequality follows from $\left\|F(z) + B_I^\top \lambda\right\| \geq \phi_I(z) \geq \delta_I$.

By combining (34) and (35), it holds that there exists a positive constant $c_I > 0$ such that for any $z \in \mathcal{P}_I$ and $\lambda \in \mathbb{R}_+^{|I|}$:

$$\text{dist}(z, S) \leq c_I \left\|F(z) + B_I^\top \lambda\right\|.$$

There are only finitely many such index sets, so we can define:

$$H := \max_{\substack{I \subseteq [p] \\ \mathcal{P}_I \neq \emptyset}} c_I > 0.$$

Since $\mathcal{P} = \bigcup_{\substack{I \subseteq [p] \\ \mathcal{P}_I \neq \emptyset}} \mathcal{P}_I$, we have for any $z \in \mathcal{P}$ and $\lambda \in \mathbb{R}_+^{|I(z)|}$:

$$\text{dist}(z, S) \leq H \left\|F(z) + B_{I(z)}^\top \lambda\right\|. \tag{36}$$

From (32), $\mathcal{N}_\mathcal{P}(z)$ is finitely generated, and hence closed (Rockafellar, 1997, Theorem 19.1). Since $\mathcal{N}_\mathcal{P}(z)$ is then a nonempty closed set, in finite dimensions the minimum $\min_{v \in \mathcal{N}_\mathcal{P}(z)} \|F(z) + v\|$ is attained. Let $v^*(z)$ be a minimizer. By (32), we can choose $\lambda^*(z) \in \mathbb{R}_+^{|I(z)|}$ such that $v^*(z) = B_{I(z)}^\top \lambda^*(z)$. Therefore,

$$\left\|F(z) + B_{I(z)}^\top \lambda^*(z)\right\| = \min_{v \in \mathcal{N}_\mathcal{P}(z)} \|F(z) + v\|.$$

Applying (36) with $\lambda = \lambda^*(z)$ yields:

$$\text{dist}(z, S) \leq H \left\|F(z) + B_{I(z)}^\top \lambda^*(z)\right\| = H \min_{v \in \mathcal{N}_\mathcal{P}(z)} \|F(z) + v\|.$$

$\square$

### G.4. Proof of Lemma G.3

*Proof of Lemma G.3.* By the first-order optimality condition for (4) and (5), $z \in \mathcal{Z}^\mu$ is equivalent to

$$\langle F_\mu^x(z), z - z' \rangle \leq 0, \ \forall z' \in \mathcal{Z},$$

which by the definition of the normal cone is equivalent to $-F_\mu^x(z) \in \mathcal{N}_\mathcal{Z}(z)$, i.e., $0 \in F_\mu^x(z) + \mathcal{N}_\mathcal{Z}(z)$. Therefore, $\mathcal{Z}^\mu = \{z \in \mathcal{Z} \mid 0 \in F_\mu^x(z) + \mathcal{N}_\mathcal{Z}(z)\}$. $\square$

## H. Proofs for Theorem 4.3

Throughout this section, we let $D := \max_{z,z' \in \mathcal{Z}} \|z - z'\|$ denote the diameter of $\mathcal{Z}$.

### H.1. Proof of Theorem 4.3

*Proof of Theorem 4.3.* First, we show that the number of perturbed games that must be solved is constant and does not depend on the target accuracy $\varepsilon$:

**Lemma H.1.** *For any target accuracy $\varepsilon > 0$, Algorithm 1 with an initial perturbation strength $\mu_{\mathrm{init}} > 0$ terminates after at most $K := \max\left(1, \lceil \frac{\ln(\mu_{\mathrm{init}} \max_{x \in \mathcal{X}} \|x\|/\alpha)}{\ln 2} \rceil + 2\right)$ episodes, and outputs a strategy profile $(x, y)$ satisfying* $\mathrm{NashConv}(x, y) \leq \varepsilon$.

By Lemma H.1, it suffices to upper bound the iteration complexity for each episode. We thus derive the following iteration complexity for solving asymmetrically perturbed games:

**Lemma H.2.** *Let us consider an arbitrary $\mu > 0$, and assume that $\eta \leq \frac{\mu}{\mu^2 + \|A\|^2}$. There exists a positive constant $\beta_\mu$ depending on $A, \mathcal{X}, \mathcal{Y}$, and $\mu$, independent of $\eta$ such that: for any $\delta > 0$,*

* ASYMP-GDA$_x(x^1, y^1, \eta, \mu, \delta)$ *in Algorithm 1 outputs the strategy profile $z = (x, y)$ satisfying*

$$\max_{z' \in \mathcal{Z}} \langle F_\mu^x(z), z - z' \rangle \leq \delta$$

  *within at most*

  $$\frac{2 \left(\ln D + \ln \left(3\mu \max_{z' \in \mathcal{Z}} \|z'\| + \sqrt{m} + \sqrt{n}\right) + \ln(1/\delta)\right)}{\ln \left(1 + \eta^2/(2\beta_\mu^2)\right)}$$

  *inner-iterations.*

* ASYMP-GDA$_y(x^1, y^1, \eta, \mu, \delta)$ *in Algorithm 1 outputs the strategy profile $z = (x, y)$ satisfying*

$$\max_{z' \in \mathcal{Z}} \langle F_\mu^y(z), z - z' \rangle \leq \delta$$

  *within at most*

  $$\frac{2 \left(\ln D + \ln \left(3\mu \max_{z' \in \mathcal{Z}} \|z'\| + \sqrt{m} + \sqrt{n}\right) + \ln(1/\delta)\right)}{\ln \left(1 + \eta^2/(2\beta_\mu^2)\right)}$$

  *inner-iterations.*

By Lemma H.2, for any $k \geq 1$, we obtain the strategy profiles $\hat{z}_x^{\mu_k} := (\hat{x}_x^{\mu_k}, \hat{y}_x^{\mu_k})$ and $\hat{z}_y^{\mu_k} := (\hat{x}_y^{\mu_k}, \hat{y}_y^{\mu_k})$ satisfying:

$$\max_{z' \in \mathcal{Z}} \langle F_\mu^x(\hat{z}_x^{\mu_k}), \hat{z}_x^{\mu_k} - z' \rangle \leq \frac{\mu_k \varepsilon^2}{2m},$$

$$\max_{z' \in \mathcal{Z}} \langle F_\mu^y(\hat{z}_y^{\mu_k}), \hat{z}_y^{\mu_k} - z' \rangle \leq \frac{\mu_k \varepsilon^2}{2n},$$

after at most $\frac{4\left(\ln\left(3\mu_k \max_{z' \in \mathcal{Z}} \|z'\| + \sqrt{m} + \sqrt{n}\right) + \ln(m+n) + \ln(2D/\mu_k) + 2\ln(1/\varepsilon)\right)}{\ln\left(1 + \eta_k^2/(2\beta_{\mu_k}^2)\right)}$ strategy updates. Therefore, the total iteration complexity is bounded by:

$$\sum_{k=1}^{K} \frac{4 \left(\ln \left(3\mu_k \max_{z' \in \mathcal{Z}} \|z'\| + \sqrt{m} + \sqrt{n}\right) + \ln (m + n) + \ln(2D/\mu_k) + 2\ln(1/\varepsilon)\right)}{\ln \left(1 + \eta_k^2/(2\beta_{\mu_k}^2)\right)}$$

$$\leq K \frac{4 \left(\ln \left(3\mu_{\mathrm{init}} \max_{z' \in \mathcal{Z}} \|z'\| + \sqrt{m} + \sqrt{n}\right) + \ln (m + n) + \ln(2D/\min_{k \in [K]} \mu_k) + 2\ln(1/\varepsilon)\right)}{\ln \left(1 + \min_{k \in [K]} \eta_k^2/(2\max_{k \in [K]} \beta_{\mu_k}^2)\right)}. \tag{37}$$

The number of episodes $K = \max\left(1, \lceil \frac{\ln(\mu_{\mathrm{init}} \max_{z \in \mathcal{Z}} \|z\|/\alpha)}{\ln 2} \rceil + 2\right)$ is finite and independent of $\varepsilon$. Since $\{\mu_k\}_{k \in [K]}$ is decreasing, we have $\mu_k \geq \mu_K$ for all $k \in [K]$, and from the definition of $K$:

$$\mu_K = \mu_{\mathrm{init}}/2^{K-1} \geq \min\left(\mu_{\mathrm{init}}, \frac{\alpha}{4\max_{z' \in \mathcal{Z}} \|z'\|}\right).$$

Similarly, since $\{\eta_k\}_{k \in [K]}$ is non-increasing, $\eta_k \geq \eta_K$ for all $k \in [K]$, and from the update rule $\eta_k = \min(\eta_{k-1}, \mu_k/(\mu_k^2 + mn))$ together with $\mu_k \in [\mu_K, \mu_{\text{init}}]$:

$$\eta_K \geq \min\left(\eta_{\text{init}}, \frac{\mu_K}{\mu_{\text{init}}^2 + mn}\right) \geq \min\left(\eta_{\text{init}}, \frac{\mu_{\text{init}}}{\mu_{\text{init}}^2 + mn}, \frac{\alpha}{4 \max_{z' \in \mathcal{Z}} \|z'\|(\mu_{\text{init}}^2 + mn)}\right).$$

Furthermore, $\{\beta_{\mu_k}\}_{k \in [K]}$ is a finite set of positive constants independent of $\varepsilon$, so there exists a positive constant $C_\beta$ independent of $\varepsilon$ such that $\max_{k \in [K]} \beta_{\mu_k} \leq C_\beta$. Combining these bounds, there exist positive constants $C_1, C_2, C_3, C_4$ independent of $\varepsilon$ such that the right-hand side of (37) is upper-bounded by:

$$\frac{C_1(C_2 + C_3 \ln(1/\varepsilon))}{C_4}.$$

Therefore, we conclude that we can obtain an $\varepsilon$-equilibrium in the original game within $\mathcal{O}(\ln(1/\varepsilon))$ strategy updates. $\qquad\square$

### H.2. Proof of Lemma H.1

*Proof of Lemma H.1.* We first show that, for sufficiently large $k$, the equilibrium strategy of the perturbed game exactly matches an equilibrium strategy in the original game:

**Lemma H.3.** *Let us define* $\mu_k := \frac{\mu_{\text{init}}}{2^{k-1}}$, $x^{\mu_k} := \arg\min_{x \in \mathcal{X}} \max_{y \in \mathcal{Y}} \{x^\top A y + \frac{\mu_k}{2}\|x\|^2\}$, *and* $y^{\mu_k} := \arg\max_{y \in \mathcal{Y}} \min_{x \in \mathcal{X}} \{x^\top A y - \frac{\mu_k}{2}\|y\|^2\}$. *Then, for any* $k > \lceil \frac{\ln(\mu_{\text{init}} \max_{z \in \mathcal{Z}} \|z\|/\alpha)}{\ln 2} \rceil + 1$, *we have* $x^{\mu_k} \in \mathcal{X}^*$ *and* $y^{\mu_k} \in \mathcal{Y}^*$.

We also prove that, in the regime where $x^\mu \in \mathcal{X}^*$ and $y^\mu \in \mathcal{Y}^*$, the NashConv can be upper bounded by the suboptimality gap in the perturbed game:

**Lemma H.4.** *Assume that* $\mu \in (0, \frac{\alpha}{\max_{x \in \mathcal{X}} \|x\| + \max_{y \in \mathcal{Y}} \|y\|})$. *If* $z = (x, y) \in \mathcal{Z}$ *satisfies for an arbitrary non-negative value* $\varepsilon \geq 0$:

$$\max_{z' \in \mathcal{Z}} \langle F_\mu^x(z), z - z' \rangle \leq \frac{\mu}{2m}\varepsilon^2,$$

*then:*

$$\max_{y' \in \mathcal{Y}} x^\top A y' - v^* \leq \varepsilon.$$

*Furthermore, if* $z = (x, y) \in \mathcal{Z}$ *satisfies:*

$$\max_{z' \in \mathcal{Z}} \langle F_\mu^y(z), z - z' \rangle \leq \frac{\mu}{2n}\varepsilon^2,$$

*then:*

$$v^* - \min_{x' \in \mathcal{X}} (x')^\top A y \leq \varepsilon.$$

Let us define $K := \max\left(1, \lceil \frac{\ln(\mu_{\text{init}} \max_{z \in \mathcal{Z}} \|z\|/\alpha)}{\ln 2} \rceil + 2\right)$. By combining Lemmas H.3 and H.4, whenever ASYMP-GDA$_x$ and ASYMP-GDA$_y$ output $\frac{\mu_k \varepsilon^2}{2\|A\|^2 \max_{z' \in \mathcal{Z}} \|z'\|^2}$-equilibrium in perturbed games for any $k \geq 1$, Algorithm 1 terminates after at most $K$ outer-iterations for achieving the strategy profile $(x, y)$ satisfying NashConv$(x, y) \leq \varepsilon$. $\qquad\square$

### H.3. Proof of Lemma H.2

*Proof of Lemma H.2.* Let us denote the equilibrium for player x's perturbed game as $x_x^\mu := \arg\min_{x \in \mathcal{X}} \max_{y \in \mathcal{Y}} \{x^\top A y + \frac{\mu}{2}\|x\|^2\}$ and $\mathcal{Y}_x^\mu = \arg\max_{y \in \mathcal{Y}} \min_{x \in \mathcal{X}} \{x^\top A y + \frac{\mu}{2}\|x\|^2\}$. Also, let us denote the equilibrium for player y's perturbed game as $y_y^\mu := \arg\max_{y \in \mathcal{Y}} \min_{x \in \mathcal{X}} \{x^\top A y - \frac{\mu}{2}\|y\|^2\}$ and $\mathcal{X}_y^\mu := \arg\min_{x \in \mathcal{X}} \max_{y \in \mathcal{Y}} \{x^\top A y - \frac{\mu}{2}\|y\|^2\}$,.

By applying the same proof technique as in Theorem 4.1 to a given $\mu$, under the assumption that $\eta \leq \frac{\mu}{\mu^2 + \|A\|^2}$, we have for any $t \geq 1$:

$$\text{dist}\left(z^{t+1}, \{x_x^\mu\} \times \mathcal{Y}_x^\mu\right)^2 \leq \left(\frac{1}{1 + \eta^2/(2\beta_\mu^2)}\right)^t \text{dist}\left(z^1, \{x_x^\mu\} \times \mathcal{Y}_x^\mu\right)^2, \tag{38}$$

where $\beta_\mu > 0$ is a positive constant depending on $A, \mathcal{X}, \mathcal{Y}$, and $\mu$, independent of $\eta$. Then, we can bound the gap function by the distance from the equilibrium set as follows:

**Lemma H.5.** *We have for any $z \in \mathcal{Z}$ and $z_x^\mu \in \{x_x^\mu\} \times \mathcal{Y}_x^\mu$:*

$$\max_{z' \in \mathcal{Z}} \langle F_\mu^x(z), z - z' \rangle \leq \left(3\mu \max_{z' \in \mathcal{Z}} \|z'\| + \sqrt{m} + \sqrt{n}\right) \|z - z_x^\mu\|.$$

*Moreover, we have for any $z \in \mathcal{Z}$ and $z_y^\mu \in \mathcal{X}_y^\mu \times \{y_y^\mu\}$:*

$$\max_{z' \in \mathcal{Z}} \langle F_\mu^y(z), z - z' \rangle \leq \left(3\mu \max_{z' \in \mathcal{Z}} \|z'\| + \sqrt{m} + \sqrt{n}\right) \|z - z_y^\mu\|.$$

By combining Lemma H.5 and (38), we have for any $t \geq 1$:

$$\max_{z' \in \mathcal{Z}} \langle F_\mu^x(z^{t+1}), z^{t+1} - z' \rangle$$

$$\leq \left(3\mu \max_{z' \in \mathcal{Z}} \|z'\| + \sqrt{m} + \sqrt{n}\right) \text{dist}\left(z^{t+1}, \{x_x^\mu\} \times \mathcal{Y}_x^\mu\right)$$

$$\leq \left(3\mu \max_{z' \in \mathcal{Z}} \|z'\| + \sqrt{m} + \sqrt{n}\right) \left(\frac{1}{1 + \eta^2/(2\beta_\mu^2)}\right)^{t/2} \text{dist}\left(z^1, \{x_x^\mu\} \times \mathcal{Y}_x^\mu\right)$$

$$\leq D \left(3\mu \max_{z' \in \mathcal{Z}} \|z'\| + \sqrt{m} + \sqrt{n}\right) \left(\frac{1}{1 + \eta^2/(2\beta_\mu^2)}\right)^{t/2}.$$

Thus, ASYMP-GDA$_x$ returns the strategy profile $z = (x, y)$ satisfying $\max_{z' \in \mathcal{Z}} \langle F_\mu^x(z), z - z' \rangle \leq \delta$ within at most $\frac{2\left(\ln D + \ln\left(3\mu \max_{z' \in \mathcal{Z}} \|z'\| + \sqrt{m} + \sqrt{n}\right) + \ln(1/\delta)\right)}{\ln\left(1 + \eta^2/(2\beta_\mu^2)\right)}$ iterations.

By a similar argument, we can show that ASYMP-GDA$_y$ returns the strategy profile $z = (x, y)$ satisfying $\max_{z' \in \mathcal{Z}} \langle F_\mu^y(z), z - z' \rangle \leq \delta$ within at most $\frac{2\left(\ln D + \ln\left(3\mu \max_{z' \in \mathcal{Z}} \|z'\| + \sqrt{m} + \sqrt{n}\right) + \ln(1/\delta)\right)}{\ln\left(1 + \eta^2/(2\beta_\mu^2)\right)}$ iterations. $\qquad \square$

## H.4. Proof of Lemma H.3

*Proof of Lemma H.3.* When $k > \lceil \frac{\ln(\mu_{\text{init}} \max_{z \in \mathcal{Z}} \|z\|/\alpha)}{\ln 2} \rceil + 1$, we have $\mu_k = \frac{\mu_{\text{init}}}{2^{k-1}} < \frac{\alpha}{\max_{z \in \mathcal{Z}} \|z\|} \leq \frac{\alpha}{\max_{x \in \mathcal{X}} \|x\|}$. Hence, the assumption that $\mu_k < \frac{\alpha}{\max_{x \in \mathcal{X}} \|x\|}$ in Corollary 3.2 satisfies when $k > \lceil \frac{\ln(\mu_{\text{init}} \max_{x \in \mathcal{X}} \|x\|/\alpha)}{\ln 2} \rceil + 1$. Therefore, we have $x^{\mu_k} \in \mathcal{X}^*$ for any $k > \lceil \frac{\ln(\mu_{\text{init}} \max_{z \in \mathcal{Z}} \|z\|/\alpha)}{\ln 2} \rceil + 1$. By a similar argument, we have $y^{\mu_k} \in \mathcal{Y}^*$ for any $k > \lceil \frac{\ln(\mu_{\text{init}} \max_{z \in \mathcal{Z}} \|z\|/\alpha)}{\ln 2} \rceil + 1$. $\qquad \square$

## H.5. Proof of Lemma H.4

*Proof of Lemma H.4.* We have for any $x \in \mathcal{X}$ and $x^* \in \mathcal{X}^*$:

$$\max_{y' \in \mathcal{Y}} x^\top A y' - v^* = \max_{y' \in \mathcal{Y}} \left((x - x^*)^\top A y' + (x^*)^\top A y'\right) - v^*$$

$$\leq \max_{y' \in \mathcal{Y}} (x - x^*)^\top A y' + \max_{y' \in \mathcal{Y}} (x^*)^\top A y' - v^*$$

$$= \max_{y' \in \mathcal{Y}} (x - x^*)^\top A y'$$

$$\leq \sqrt{m} \|x - x^*\|.$$

From Corollary 3.2, under the assumption that $\mu \in (0, \frac{\alpha}{\max_{x \in \mathcal{X}} \|x\|})$, it holds that $x^\mu \in \mathcal{X}^*$. Hence, we get for any $x \in \mathcal{X}$:

$$\max_{y' \in \mathcal{Y}} x^\top A y' - v^* \leq \sqrt{m} \|x - x^\mu\|. \tag{39}$$

On the other hand, since the function $g_{\text{asym}}^\mu(x) = \max_{y' \in \mathcal{Y}} x^\top A y' + \frac{\mu}{2} \|x\|^2$ is $\mu$-strongly convex, we have for any subgradient $s \in \partial g_{\text{asym}}^\mu(x')$:

$$\forall x \in \mathcal{X}, \ \frac{\mu}{2} \|x - x'\|^2 \leq \langle s, x' - x \rangle + g_{\text{asym}}^\mu(x) - g_{\text{asym}}^\mu(x').$$

From the first-order optimality condition for $x^\mu$, there must be a subgradient $s^* \in \partial g_{\text{asym}}^\mu(x^\mu)$ such that:

$$\forall x \in \mathcal{X}, \ \langle s^*, x^\mu - x \rangle \leq 0.$$

Thus, setting $x' = x^\mu$, we have:

$$\forall x \in \mathcal{X}, \ \frac{\mu}{2} \|x - x^\mu\|^2 \leq g_{\text{asym}}^\mu(x) - g_{\text{asym}}^\mu(x^\mu).$$

The right-hand side in the above inequality can be upper bounded as follows: for any $y \in \mathcal{Y}$,

$$
\begin{aligned}
& g_{\text{asym}}^\mu(x) - g_{\text{asym}}^\mu(x^\mu) \\
&= \max_{y' \in \mathcal{Y}} x^\top A y' + \frac{\mu}{2} \|x\|^2 - \min_{x' \in \mathcal{X}} \max_{y' \in \mathcal{Y}} \left( (x')^\top A y' + \frac{\mu}{2} \|x'\|^2 \right) \\
&= \max_{y' \in \mathcal{Y}} x^\top A y' + \frac{\mu}{2} \|x\|^2 - \max_{y' \in \mathcal{Y}} \min_{x' \in \mathcal{X}} \left( (x')^\top A y' + \frac{\mu}{2} \|x'\|^2 \right) \\
&\leq \max_{y' \in \mathcal{Y}} x^\top A y' + \frac{\mu}{2} \|x\|^2 - \min_{x' \in \mathcal{X}} \left( (x')^\top A y + \frac{\mu}{2} \|x'\|^2 \right) \\
&= \max_{y' \in \mathcal{Y}} \langle -A^\top x, y - y' \rangle + x^\top A y + \frac{\mu}{2} \|x\|^2 - \min_{x' \in \mathcal{X}} \left( (x')^\top A y + \frac{\mu}{2} \|x'\|^2 \right) \\
&\leq \max_{y' \in \mathcal{Y}} \langle -A^\top x, y - y' \rangle + \langle Ay + \mu x, x - x^\dagger \rangle \\
&\leq \max_{x' \in \mathcal{X}} \langle Ay + \mu x, x - x' \rangle + \max_{y' \in \mathcal{Y}} \langle -A^\top x, y - y' \rangle \\
&= \max_{z' \in \mathcal{Z}} \langle F_\mu^x(z), z - z' \rangle.
\end{aligned}
$$

Hence, we obtain:

$$\forall z = (x, y) \in \mathcal{Z}, \ \frac{\mu}{2} \|x - x^\mu\|^2 \leq \max_{z' \in \mathcal{Z}} \langle F_\mu^x(z), z - z' \rangle. \tag{40}$$

By combining (39) and (40), we finally obtain:

$$\max_{y' \in \mathcal{Y}} x^\top A y' - v^* \leq \sqrt{\frac{2m}{\mu}} \sqrt{\max_{z' \in \mathcal{Z}} \langle F_\mu^x(z), z - z' \rangle}.$$

Therefore,

$$\max_{z' \in \mathcal{Z}} \langle F_\mu^x(z), z - z' \rangle \leq \frac{\mu}{2m} \varepsilon^2 \Rightarrow \max_{y' \in \mathcal{Y}} x^\top A y' - v^* \leq \varepsilon.$$

By a similar argument, under the assumption that $\mu \in (0, \frac{\alpha}{\max_{y \in \mathcal{Y}} \|y\|})$, we have:

$$\max_{z' \in \mathcal{Z}} \langle F_\mu^y(z), z - z' \rangle \leq \frac{\mu}{2n} \varepsilon^2 \Rightarrow v^* - \min_{x' \in \mathcal{X}} (x')^\top A y \leq \varepsilon.$$

$\square$

## H.6. Proof of Lemma H.5

*Proof of Lemma H.5.* For any $z = (x, y), \hat{z} = (\hat{x}, \hat{y}), z' = (x', y') \in \mathcal{Z}$, expanding the inner products yields:

$$
\begin{aligned}
&\langle F_\mu^x(z), z - z' \rangle - \langle F_\mu^x(\hat{z}), \hat{z} - z' \rangle \\
&= -\langle Ay, x' \rangle + \mu\langle x, x - x' \rangle + \langle A^\top x, y' \rangle + \langle A\hat{y}, x' \rangle - \mu\langle \hat{x}, \hat{x} - x' \rangle - \langle A^\top \hat{x}, y' \rangle \\
&= -\langle A(y - \hat{y}), x' \rangle + \mu\langle x, x - x' \rangle - \mu\langle \hat{x}, \hat{x} - x' \rangle + \langle A^\top(x - \hat{x}), y' \rangle \\
&= -\langle A(y - \hat{y}), x' \rangle + \mu\langle x - \hat{x}, x - x' \rangle + \mu\langle \hat{x}, x - \hat{x} \rangle + \langle A^\top(x - \hat{x}), y' \rangle \\
&\leq \|A^\top x'\|\|y - \hat{y}\| + \mu\|x - x'\|\|x - \hat{x}\| + \mu\|\hat{x}\|\|x - \hat{x}\| + \|Ay'\|\|x - \hat{x}\|.
\end{aligned}
$$

Hence, we get for any $z, \hat{z} \in \mathcal{Z}$:

$$
\begin{aligned}
&\max_{z' \in \mathcal{Z}}\langle F_\mu^x(z), z - z' \rangle - \max_{z' \in \mathcal{Z}}\langle F_\mu^x(\hat{z}), \hat{z} - z' \rangle \\
&\leq \max_{z' \in \mathcal{Z}}\left(\langle F_\mu^x(z), z - z' \rangle - \langle F_\mu^x(\hat{z}), \hat{z} - z' \rangle\right) \\
&\leq \max_{z' \in \mathcal{Z}}\left(\|A^\top x'\|\|y - \hat{y}\| + \mu\|x - x'\|\|x - \hat{x}\| + \mu\|\hat{x}\|\|x - \hat{x}\| + \|Ay'\|\|x - \hat{x}\|\right) \\
&\leq \max_{x' \in \mathcal{X}}\|A^\top x'\|\|y - \hat{y}\| + 3\mu\max_{x' \in \mathcal{X}}\|x'\|\|x - \hat{x}\| + \max_{y' \in \mathcal{Y}}\|Ay'\|\|x - \hat{x}\| \\
&\leq \left(3\mu\max_{x' \in \mathcal{X}}\|x'\| + \max_{x' \in \mathcal{X}}\|A^\top x'\| + \max_{y' \in \mathcal{Y}}\|Ay'\|\right)\|z - \hat{z}\| \\
&\leq \left(3\mu\max_{z' \in \mathcal{Z}}\|z'\| + \sqrt{m} + \sqrt{n}\right)\|z - \hat{z}\|.
\end{aligned}
$$

Here, from the first-order optimality condition for $x^\mu$ and $y^\mu \in \mathcal{Y}^\mu$, writing $z^\mu := (x^\mu, y^\mu)$, we have:

$$
\max_{z' \in \mathcal{Z}}\langle F_\mu^x(z^\mu), z^\mu - z' \rangle \leq 0.
$$

By combining these inequalities, we obtain for any $z \in \mathcal{Z}$ and $z^\mu \in \mathcal{Z}^\mu$:

$$
\max_{z' \in \mathcal{Z}}\langle F_\mu^x(z), z - z' \rangle \leq \left(3\mu\max_{z' \in \mathcal{Z}}\|z'\| + \sqrt{m} + \sqrt{n}\right)\|z - z^\mu\|.
$$

By a similar argument, we have for any $z \in \mathcal{Z}$ and $z^\mu \in \mathcal{Z}^\mu$:

$$
\max_{z' \in \mathcal{Z}}\langle F_\mu^y(z), z - z' \rangle \leq \left(3\mu\max_{z' \in \mathcal{Z}}\|z'\| + \sqrt{m} + \sqrt{n}\right)\|z - z^\mu\|.
$$

$\square$

# I. Proof of Theorem B.1

*Proof of Theorem B.1.* First, we prove that $y^\mu \neq y^*$ under the assumption that $\mu \neq \frac{v^* - \left(\frac{1_m}{m}\right)^\top A\left(\frac{1_n}{n}\right)}{\left(\|y^*\|^2 - \frac{1}{n}\right)}$ by contradiction. We assume that $y^\mu = y^*$. Since $(x^*, y^*)$ is in the interior of $\Delta^m \times \Delta^n$, we have:

$$
\begin{aligned}
(A^\top x^*)_i &= v^*, \ \forall i \in [m] \\
(Ay^*)_i &= v^*, \ \forall i \in [n].
\end{aligned}
\tag{41}
$$

Then, from (41), we have for any $x \in \Delta^m$:

$$
x^\top Ay^\mu + \frac{\mu}{2}\|x\|^2 = x^\top Ay^* + \frac{\mu}{2}\|x\|^2 = v^* + \frac{\mu}{2}\|x\|^2.
\tag{42}
$$

On the other hand,

$$\left(\frac{\mathbf{1}_m}{m}\right)^\top A^\top y^\mu + \frac{\mu}{2}\left\|\frac{\mathbf{1}_m}{m}\right\|^2 = \left(\frac{\mathbf{1}_m}{m}\right)^\top A^\top y^* + \frac{\mu}{2}\left\|\frac{\mathbf{1}_m}{m}\right\|^2 = v^* + \frac{\mu}{2}\left\|\frac{\mathbf{1}_m}{m}\right\|^2. \tag{43}$$

By combining (42) and (43), we have for any $x \in \Delta^m$:

$$x^\top A y^\mu + \frac{\mu}{2}\|x\|^2 \geq \left(\frac{\mathbf{1}_m}{m}\right)^\top A^\top y^\mu + \frac{\mu}{2}\left\|\frac{\mathbf{1}_m}{m}\right\|^2.$$

Hence, from the property of the player $x$'s equilibrium strategy in the perturbed game, $x^\mu$ must satisfy $x^\mu = \frac{\mathbf{1}_m}{m}$.

On the other hand, from the property of the player $y$'s equilibrium strategy $y^\mu$ in the perturbed game, $y^\mu$ is an optimal solution of the following optimization problem:

$$\max_{y \in \Delta^n} \left\{ (x^\mu)^\top A y - \frac{\mu}{2}\|y\|^2 \right\}.$$

Let us define the following Lagrangian function $L(y, \kappa, \lambda)$ as:

$$L(y, \kappa, \lambda) = (x^\mu)^\top A y - \frac{\mu}{2}\|y\|^2 - \sum_{i=1}^n \kappa_i g_i(y) - \lambda h(y),$$

where $g_i(y) = -y_i$ and $h(y) = \sum_{i=1}^n y_i - 1$. Then, from the KKT conditions, we get the stationarity:

$$A^\top x^\mu - \mu y^\mu - \sum_{i=1}^n \kappa_i \nabla g_i(y^\mu) - \lambda \nabla h(y^\mu) = \mathbf{0}_n, \tag{44}$$

and the complementary slackness:

$$\forall i \in [n], \ \kappa_i g_i(y^\mu) = 0. \tag{45}$$

Since $y^\mu = y^*$ and $y^*$ is in the interior of $\Delta^n$, we have $g(y^\mu) = -y_i^\mu < 0$ for all $i \in [n]$. Thus, from (45), we have $\kappa_i = 0$ for all $i \in [n]$. Substituting this into (44), we obtain:

$$A^\top x^\mu - \mu y^\mu - \lambda \nabla h(y^\mu) = A^\top x^\mu - \mu y^\mu - \lambda \mathbf{1}_n = \mathbf{0}_n. \tag{46}$$

Hence, we have:

$$\lambda = \frac{\mathbf{1}_n^\top A^\top x^\mu - \mu}{n}. \tag{47}$$

Putting (47) into (46) yields:

$$y^\mu = \frac{1}{\mu}\left(A^\top x^\mu - \frac{\mathbf{1}_n^\top A^\top x^\mu - \mu}{n}\mathbf{1}_n\right) = \frac{1}{\mu}\left(A^\top \frac{\mathbf{1}_m}{m} - \frac{\mathbf{1}_n^\top A^\top \frac{\mathbf{1}_m}{m} - \mu}{n}\mathbf{1}_n\right),$$

where the second equality follows from $x^\mu = \frac{\mathbf{1}_m}{m}$. Multiplying this by $\frac{\mathbf{1}_m^\top}{m}A$, we have:

$$v^* = \frac{1}{\mu}\left(\frac{1}{m^2}\|A^\top \mathbf{1}_m\|^2 - \frac{1}{m^2 n}(\mathbf{1}_n^\top A^\top \mathbf{1}_m)^2 + \mu\frac{1}{mn}\mathbf{1}_m^\top A \mathbf{1}_n\right), \tag{48}$$

where we used the assumption that $y^\mu = y^*$ and (41). Here, denoting the $n \times n$ identity matrix by $\mathbb{I}$, we have:

$$\frac{2}{m^2} \left\| A^\top \mathbf{1}_m \right\|^2 - \frac{1}{m^2 n} (\mathbf{1}_n^\top A^\top \mathbf{1}_m)^2 + \mu \frac{1}{mn} \mathbf{1}_m^\top A \mathbf{1}_n$$

$$= \left\| v^* \mathbf{1}_n + A^\top \frac{\mathbf{1}_m}{m} - v^* \mathbf{1}_n \right\|^2 - \frac{1}{n} \left( \mathbf{1}_n^\top \left( v^* \mathbf{1}_n + A^\top \frac{\mathbf{1}_m}{m} - v^* \mathbf{1}_n \right) \right)^2 + \frac{\mu}{n} \mathbf{1}_n^\top \left( v^* \mathbf{1}_n + A^\top \frac{\mathbf{1}_m}{m} - v^* \mathbf{1}_n \right)$$

$$= n(v^*)^2 + \left\| A^\top \frac{\mathbf{1}_m}{m} - v^* \mathbf{1}_n \right\|^2 + 2v^* \mathbf{1}_n^\top \left( A^\top \frac{\mathbf{1}_m}{m} - v^* \mathbf{1}_n \right)$$

$$\quad - n(v^*)^2 - \frac{1}{n} \left( \mathbf{1}_n^\top \left( A^\top \frac{\mathbf{1}_m}{m} - v^* \mathbf{1}_n \right) \right)^2 - 2v^* \mathbf{1}_n^\top \left( A^\top \frac{\mathbf{1}_m}{m} - v^* \mathbf{1}_n \right) + \mu v^* + \frac{\mu}{n} \mathbf{1}_n^\top \left( A^\top \frac{\mathbf{1}_m}{m} - v^* \mathbf{1}_n \right)$$

$$= \mu v^* + \left\| A^\top \frac{\mathbf{1}_m}{m} - v^* \mathbf{1}_n \right\|^2 - \frac{1}{n} \left( \mathbf{1}_n^\top \left( A^\top \frac{\mathbf{1}_m}{m} - v^* \mathbf{1}_n \right) \right)^2 + \frac{\mu}{n} \mathbf{1}_n^\top \left( A^\top \frac{\mathbf{1}_m}{m} - v^* \mathbf{1}_n \right)$$

$$= \mu v^* + \left( A^\top \frac{\mathbf{1}_m}{m} - v^* \mathbf{1}_n \right)^\top \left( \mathbb{I} - \frac{1}{n} \mathbf{1}_n \mathbf{1}_n^\top \right) \left( A^\top \frac{\mathbf{1}_m}{m} - v^* \mathbf{1}_n \right) + \frac{\mu}{n} \mathbf{1}_n^\top \left( A^\top \frac{\mathbf{1}_m}{m} - v^* \mathbf{1}_n \right). \tag{49}$$

Here, since $A^\top \frac{\mathbf{1}_m}{m} = \frac{\mathbf{1}_n^\top A^\top x^\mu - \mu}{n} \mathbf{1}_n + \mu y^\mu = \frac{\frac{1}{m} \mathbf{1}_m^\top A \mathbf{1}_n - \mu}{n} \mathbf{1}_n + \mu y^*$ from (46) and (47), we get:

$$\left( A^\top \frac{\mathbf{1}_m}{m} - v^* \mathbf{1}_n \right)^\top \left( \mathbb{I} - \frac{1}{n} \mathbf{1}_n \mathbf{1}_n^\top \right) \left( A^\top \frac{\mathbf{1}_m}{m} - v^* \mathbf{1}_n \right) + \frac{\mu}{n} \mathbf{1}_n^\top \left( A^\top \frac{\mathbf{1}_m}{m} - v^* \mathbf{1}_n \right)$$

$$= \left( \left( \frac{\frac{1}{m} \mathbf{1}_m^\top A \mathbf{1}_n - \mu}{n} - v^* \right) \mathbf{1}_n + \mu y^* \right)^\top \left( \mathbb{I} - \frac{1}{n} \mathbf{1}_n \mathbf{1}_n^\top \right) \left( \left( \frac{\frac{1}{m} \mathbf{1}_m^\top A \mathbf{1}_n - \mu}{n} - v^* \right) \mathbf{1}_n + \mu y^* \right)$$

$$\quad + \frac{\mu}{n} \mathbf{1}_n^\top \left( \left( \frac{\frac{1}{m} \mathbf{1}_m^\top A \mathbf{1}_n - \mu}{n} - v^* \right) \mathbf{1}_n + \mu y^* \right)$$

$$= \mu \left( \frac{\frac{1}{m} \mathbf{1}_m^\top A \mathbf{1}_n - \mu}{n} - v^* \right) + \frac{\mu^2}{n} + \mu \left( \frac{\frac{1}{m} \mathbf{1}_m^\top A \mathbf{1}_n - \mu}{n} - v^* \right) \mathbf{1}_n^\top \left( \mathbb{I} - \frac{1}{n} \mathbf{1}_n \mathbf{1}_n^\top \right) y^*$$

$$\quad + \mu \left( \frac{\frac{1}{m} \mathbf{1}_m^\top A \mathbf{1}_n - \mu}{n} - v^* \right) (y^*)^\top \left( \mathbb{I} - \frac{1}{n} \mathbf{1}_n \mathbf{1}_n^\top \right) \mathbf{1}_n + \mu^2 (y^*)^\top \left( \mathbb{I} - \frac{1}{n} \mathbf{1}_n \mathbf{1}_n^\top \right) y^*$$

$$= \mu \left( \frac{\frac{1}{m} \mathbf{1}_m^\top A \mathbf{1}_n - \mu}{n} - v^* \right) + \mu^2 \|y^*\|^2$$

$$= \mu \left( \mu \left( \|y^*\|^2 - \frac{1}{n} \right) + \left( \frac{\mathbf{1}_m}{m} \right)^\top A \left( \frac{\mathbf{1}_n}{n} \right) - v^* \right). \tag{50}$$

By combining (48), (49), and (50), we have:

$$\mu \left( \|y^*\|^2 - \frac{1}{n} \right) + \left( \frac{\mathbf{1}_m}{m} \right)^\top A \left( \frac{\mathbf{1}_n}{n} \right) - v^* = 0.$$

Therefore, if $y^\mu = y^*$, then $\mu$ must satisfy:

$$\mu = \frac{v^* - \left( \frac{\mathbf{1}_m}{m} \right)^\top A \left( \frac{\mathbf{1}_n}{n} \right)}{\left( \|y^*\|^2 - \frac{1}{n} \right)},$$

and this is equivalent to:

$$\mu \neq \frac{v^* - \left( \frac{\mathbf{1}_m}{m} \right)^\top A \left( \frac{\mathbf{1}_n}{n} \right)}{\left( \|y^*\|^2 - \frac{1}{n} \right)} \Rightarrow y^\mu \neq y^*.$$

By a similar argument, in terms of player $x$, we can conclude that:

$$\mu \neq \frac{\left(\frac{\mathbf{1}_m}{m}\right)^\top A \left(\frac{\mathbf{1}_n}{n}\right) - v^*}{\left(\|x^*\|^2 - \frac{1}{m}\right)} \Rightarrow x^\mu \neq x^*.$$

$\square$

## J. Proof of Theorem C.1

*Proof of Theorem C.1.* First, we prove that $y^\mu \neq y^*$ under the assumption that $\mu \neq \frac{v^* - \left(\frac{\mathbf{1}_m}{m}\right)^\top A \left(\frac{\mathbf{1}_n}{n}\right)}{\|y^*\|^2 - \frac{1}{n}}$ by contradiction. We assume that $y^\mu = y^*$. From (41), we have for any $x \in \Delta^m$:

$$x^\top A y^\mu + \frac{\mu_x}{2} \|x\|^2 = x^\top A y^* + \frac{\mu_x}{2} \|x\|^2 = v^* + \frac{\mu_x}{2} \|x\|^2. \tag{51}$$

On the other hand,

$$\left(\frac{\mathbf{1}_m}{m}\right)^\top A^\top y^\mu + \frac{\mu_x}{2} \left\|\frac{\mathbf{1}_m}{m}\right\|^2 = \left(\frac{\mathbf{1}_m}{m}\right)^\top A^\top y^* + \frac{\mu_x}{2} \left\|\frac{\mathbf{1}_m}{m}\right\|^2 = v^* + \frac{\mu_x}{2} \left\|\frac{\mathbf{1}_m}{m}\right\|^2. \tag{52}$$

By combining (51) and (52), we have for any $x \in \Delta^m$:

$$x^\top A y^\mu + \frac{\mu_x}{2} \|x\|^2 \geq \left(\frac{\mathbf{1}_m}{m}\right)^\top A^\top y^\mu + \frac{\mu_x}{2} \left\|\frac{\mathbf{1}_m}{m}\right\|^2.$$

Hence, from the property of the player $x$'s equilibrium strategy in the perturbed game, $x^\mu$ must satisfy $x^\mu = \frac{\mathbf{1}_m}{m}$.

On the other hand, from the property of the player $y$'s equilibrium strategy $y^\mu$ in the perturbed game, $y^\mu$ is an optimal solution of the following optimization problem:

$$\max_{y \in \Delta^n} \left\{ (x^\mu)^\top A y - \frac{\mu_y}{2} \|y\|^2 \right\}.$$

Let us define the following Lagrangian function $L(y, \kappa, \lambda)$ as:

$$L(y, \kappa, \lambda) = (x^\mu)^\top A y - \frac{\mu_y}{2} \|y\|^2 - \sum_{i=1}^n \kappa_i g_i(y) - \lambda h(y),$$

where $g_i(y) = -y_i$ and $h(y) = \sum_{i=1}^n y_i - 1$. Then, from the KKT conditions, we get the stationarity:

$$A^\top x^\mu - \mu_y y^\mu - \sum_{i=1}^n \kappa_i \nabla g_i(y^\mu) - \lambda \nabla h(y^\mu) = \mathbf{0}_n, \tag{53}$$

and the complementary slackness:

$$\forall i \in [n], \ \kappa_i g_i(y^\mu) = 0. \tag{54}$$

Since $y^\mu = y^*$ and $y^*$ is in the interior of $\Delta^n$, we have $g(y^\mu) = -y_i^\mu < 0$ for all $i \in [n]$. Thus, from (54), we have $\kappa_i = 0$ for all $i \in [n]$. Substituting this into (53), we obtain:

$$A^\top x^\mu - \mu_y y^\mu - \lambda \nabla h(y^\mu) = A^\top x^\mu - \mu_y y^\mu - \lambda \mathbf{1}_n = \mathbf{0}_n. \tag{55}$$

Hence, we have:

$$\lambda = \frac{\mathbf{1}_n^\top A^\top x^\mu - \mu_y}{n}. \tag{56}$$

Putting (56) into (55) yields:

$$y^\mu = \frac{1}{\mu_y}\left(A^\top x^\mu - \frac{\mathbf{1}_n^\top A^\top x^\mu - \mu_y}{n}\mathbf{1}_n\right) = \frac{1}{\mu_y}\left(A^\top \frac{\mathbf{1}_m}{m} - \frac{\mathbf{1}_n^\top A^\top \frac{\mathbf{1}_m}{m} - \mu_y}{n}\mathbf{1}_n\right),$$

where the second equality follows from $x^\mu = \frac{\mathbf{1}_m}{m}$. Multiplying this by $\frac{\mathbf{1}_m^\top}{m}A$, we have:

$$v^* = \frac{1}{\mu_y}\left(\frac{1}{m^2}\left\|A^\top \mathbf{1}_m\right\|^2 - \frac{1}{m^2 n}(\mathbf{1}_n^\top A^\top \mathbf{1}_m)^2 + \mu_y \frac{1}{mn}\mathbf{1}_m^\top A \mathbf{1}_n\right), \tag{57}$$

where we used the assumption that $y^\mu = y^*$ and (41). Here, denoting the $n \times n$ identity matrix by $\mathbb{I}$, we have:

$$\frac{1}{m^2}\left\|A^\top \mathbf{1}_m\right\|^2 - \frac{1}{m^2 n}(\mathbf{1}_n^\top A^\top \mathbf{1}_m)^2 + \mu_y \frac{1}{mn}\mathbf{1}_m^\top A \mathbf{1}_n$$

$$= \left\|v^*\mathbf{1}_n + A^\top \frac{\mathbf{1}_m}{m} - v^*\mathbf{1}_n\right\|^2 - \frac{1}{n}\left(\mathbf{1}_n^\top\left(v^*\mathbf{1}_n + A^\top \frac{\mathbf{1}_m}{m} - v^*\mathbf{1}_n\right)\right)^2 + \frac{\mu_y}{n}\mathbf{1}_n^\top\left(v^*\mathbf{1}_n + A^\top \frac{\mathbf{1}_m}{m} - v^*\mathbf{1}_n\right)$$

$$= n(v^*)^2 + \left\|A^\top \frac{\mathbf{1}_m}{m} - v^*\mathbf{1}_n\right\|^2 + 2v^*\mathbf{1}_n^\top\left(A^\top \frac{\mathbf{1}_m}{m} - v^*\mathbf{1}_n\right)$$

$$\quad - n(v^*)^2 - \frac{1}{n}\left(\mathbf{1}_n^\top\left(A^\top \frac{\mathbf{1}_m}{m} - v^*\mathbf{1}_n\right)\right)^2 - 2v^*\mathbf{1}_n^\top\left(A^\top \frac{\mathbf{1}_m}{m} - v^*\mathbf{1}_n\right) + \mu_y v^* + \frac{\mu_y}{n}\mathbf{1}_n^\top\left(A^\top \frac{\mathbf{1}_m}{m} - v^*\mathbf{1}_n\right)$$

$$= \mu_y v^* + \left\|A^\top \frac{\mathbf{1}_m}{m} - v^*\mathbf{1}_n\right\|^2 - \frac{1}{n}\left(\mathbf{1}_n^\top\left(A^\top \frac{\mathbf{1}_m}{m} - v^*\mathbf{1}_n\right)\right)^2 + \frac{\mu_y}{n}\mathbf{1}_n^\top\left(A^\top \frac{\mathbf{1}_m}{m} - v^*\mathbf{1}_n\right)$$

$$= \mu_y v^* + \left(A^\top \frac{\mathbf{1}_m}{m} - v^*\mathbf{1}_n\right)^\top\left(\mathbb{I} - \frac{1}{n}\mathbf{1}_n\mathbf{1}_n^\top\right)\left(A^\top \frac{\mathbf{1}_m}{m} - v^*\mathbf{1}_n\right) + \frac{\mu_y}{n}\mathbf{1}_n^\top\left(A^\top \frac{\mathbf{1}_m}{m} - v^*\mathbf{1}_n\right). \tag{58}$$

Here, since $A^\top \frac{\mathbf{1}_m}{m} = \frac{\mathbf{1}_n^\top A^\top x^\mu - \mu_y}{n}\mathbf{1}_n + \mu_y y^\mu = \frac{\frac{1}{m}\mathbf{1}_m^\top A\mathbf{1}_n - \mu_y}{n}\mathbf{1}_n + \mu_y y^*$ from (55) and (56), we get:

$$\left(A^\top \frac{\mathbf{1}_m}{m} - v^*\mathbf{1}_n\right)^\top\left(\mathbb{I} - \frac{1}{n}\mathbf{1}_n\mathbf{1}_n^\top\right)\left(A^\top \frac{\mathbf{1}_m}{m} - v^*\mathbf{1}_n\right) + \frac{\mu_y}{n}\mathbf{1}_n^\top\left(A^\top \frac{\mathbf{1}_m}{m} - v^*\mathbf{1}_n\right)$$

$$= \left(\left(\frac{\frac{1}{m}\mathbf{1}_m^\top A\mathbf{1}_n - \mu_y}{n} - v^*\right)\mathbf{1}_n + \mu_y y^*\right)^\top\left(\mathbb{I} - \frac{1}{n}\mathbf{1}_n\mathbf{1}_n^\top\right)\left(\left(\frac{\frac{1}{m}\mathbf{1}_m^\top A\mathbf{1}_n - \mu_y}{n} - v^*\right)\mathbf{1}_n + \mu_y y^*\right)$$

$$\quad + \frac{\mu_y}{n}\mathbf{1}_n^\top\left(\left(\frac{\frac{1}{m}\mathbf{1}_m^\top A\mathbf{1}_n - \mu_y}{n} - v^*\right)\mathbf{1}_n + \mu_y y^*\right)$$

$$= \mu_y\left(\frac{\frac{1}{m}\mathbf{1}_m^\top A\mathbf{1}_n - \mu_y}{n} - v^*\right) + \frac{\mu_y^2}{n} + \mu_y\left(\frac{\frac{1}{m}\mathbf{1}_m^\top A\mathbf{1}_n - \mu_y}{n} - v^*\right)\mathbf{1}_n^\top\left(\mathbb{I} - \frac{1}{n}\mathbf{1}_n\mathbf{1}_n^\top\right)y^*$$

$$\quad + \mu_y\left(\frac{\frac{1}{m}\mathbf{1}_m^\top A\mathbf{1}_n - \mu_y}{n} - v^*\right)(y^*)^\top\left(\mathbb{I} - \frac{1}{n}\mathbf{1}_n\mathbf{1}_n^\top\right)\mathbf{1}_n + \mu_y^2(y^*)^\top\left(\mathbb{I} - \frac{1}{n}\mathbf{1}_n\mathbf{1}_n^\top\right)y^*$$

$$= \mu_y\left(\frac{\frac{1}{m}\mathbf{1}_m^\top A\mathbf{1}_n - \mu_y}{n} - v^*\right) + \mu_y^2\left\|y^*\right\|^2$$

$$= \mu_y\left(\mu_y\left(\left\|y^*\right\|^2 - \frac{1}{n}\right) + \left(\frac{\mathbf{1}_m}{m}\right)^\top A\left(\frac{\mathbf{1}_n}{n}\right) - v^*\right). \tag{59}$$

By combining (57), (58), and (59), we have:

$$\mu_y\left(\left\|y^*\right\|^2 - \frac{1}{n}\right) + \left(\frac{\mathbf{1}_m}{m}\right)^\top A\left(\frac{\mathbf{1}_n}{n}\right) - v^* = 0.$$

Therefore, if $y^\mu = y^*$, then $\mu_y$ must satisfy:

$$\mu_y = \frac{v^* - \left(\frac{\mathbf{1}_m}{m}\right)^\top A \left(\frac{\mathbf{1}_n}{n}\right)}{\left(\|y^*\|^2 - \frac{1}{n}\right)},$$

and this is equivalent to:

$$\mu_y \neq \frac{v^* - \left(\frac{\mathbf{1}_m}{m}\right)^\top A \left(\frac{\mathbf{1}_n}{n}\right)}{\left(\|y^*\|^2 - \frac{1}{n}\right)} \Rightarrow y^\mu \neq y^*.$$

By a similar argument, in terms of player $x$, we can conclude that:

$$\mu_x \neq \frac{\left(\frac{\mathbf{1}_m}{m}\right)^\top A \left(\frac{\mathbf{1}_n}{n}\right) - v^*}{\left(\|x^*\|^2 - \frac{1}{m}\right)} \Rightarrow x^\mu \neq x^*.$$

$\square$

# K. Proof of Theorem E.1

*Proof of Theorem E.1.* Let us define $a_1 := \gamma$, $a_2 := 2\gamma$, and $a_3 := 1$, and the function $g^\mu_{\mathrm{asym}} : \Delta^3 \to \mathbb{R}$:

$$g^\mu_{\mathrm{asym}}(x) := \max_{y \in \Delta^3} x^\top A_\gamma y + \frac{\mu}{2}\|x\|^2 = \max_{i \in [3]} a_i x_i + \frac{\mu}{2}\|x\|^2.$$

Since $g^\mu_{\mathrm{asym}}$ is $\mu$-strongly convex over $\Delta^3$, its minimizer $x^\mu$ is unique. Hence, $x^\mu = x^*$ if and only if $x^*$ satisfies the first-order optimality condition for $\min_{x \in \Delta^3} g^\mu_{\mathrm{asym}}(x)$.

To derive this optimality condition explicitly, we first compute the subdifferential of $g^\mu_{\mathrm{asym}}$ at $x^*$. Since $a_1 x_1^* = a_2 x_2^* = a_3 x_3^* = \frac{2\gamma}{2\gamma+3}$, all three terms in the maximum are active at $x^*$. Thus, we have:

$$\partial g^\mu_{\mathrm{asym}}(x^*) = \left\{\mu x^* + (a_1 \lambda_1, a_2 \lambda_2, a_3 \lambda_3) \mid \lambda \in \Delta^3\right\}.$$

Moreover, since $x_i^* > 0$ for all $i \in [3]$, the normal cone of $\Delta^3$ at $x^*$ is given by:

$$\mathcal{N}_{\Delta^3}(x^*) = \{\nu \mathbf{1} \mid \nu \in \mathbb{R}\}.$$

By combining these, the first-order optimality condition $0 \in \partial g^\mu_{\mathrm{asym}}(x^*) + \mathcal{N}_{\Delta^3}(x^*)$ is equivalent to the existence of $\lambda \in \Delta^3$ and $c \in \mathbb{R}$ such that:

$$\mu x^* + (a_1 \lambda_1, a_2 \lambda_2, a_3 \lambda_3) = c\mathbf{1}. \tag{60}$$

We next solve (60) together with $\sum_i \lambda_i = 1$ to find $\lambda$ explicitly. From (60), we have $\lambda_i = (c - \mu x_i^*)/a_i$ for $i \in [3]$. Substituting this into $\sum_i \lambda_i = 1$ yields:

$$c \sum_{i=1}^3 \frac{1}{a_i} - \mu \sum_{i=1}^3 \frac{x_i^*}{a_i} = 1.$$

Here, we have:

$$\sum_{i=1}^3 \frac{1}{a_i} = \frac{1}{\gamma} + \frac{1}{2\gamma} + 1 = \frac{2\gamma+3}{2\gamma}, \qquad \sum_{i=1}^3 \frac{x_i^*}{a_i} = \frac{2}{\gamma(2\gamma+3)} + \frac{1}{2\gamma(2\gamma+3)} + \frac{2\gamma}{2\gamma+3} = \frac{4\gamma^2+5}{2\gamma(2\gamma+3)}.$$

Hence, solving for $c$ yields:

$$c = \frac{2\gamma}{2\gamma+3} + \frac{\mu(4\gamma^2+5)}{(2\gamma+3)^2}.$$

Substituting this back into $\lambda_i = (c - \mu x_i^*)/a_i$, we obtain:

$$\lambda_1 = \frac{2\gamma(2\gamma + 3) - \mu(1 + 4\gamma - 4\gamma^2)}{\gamma(2\gamma + 3)^2},$$

$$\lambda_2 = \frac{2\gamma(2\gamma + 3) + \mu(4\gamma^2 - 2\gamma + 2)}{2\gamma(2\gamma + 3)^2},$$

$$\lambda_3 = \frac{2\gamma(2\gamma + 3) - \mu(6\gamma - 5)}{(2\gamma + 3)^2}.$$

By construction, $\lambda = (\lambda_1, \lambda_2, \lambda_3)$ is the unique vector that satisfies both (60) and $\sum_i \lambda_i = 1$. Therefore, $x^*$ satisfies the first-order optimality condition if and only if $\lambda \in \Delta^3$. Since $\lambda$ already satisfies $\sum_i \lambda_i = 1$ by construction, $\lambda \in \Delta^3$ is equivalent to $\lambda_i \geq 0$ for all $i \in [3]$. Hence, it remains to show that $\lambda_i \geq 0$ for all $i \in [3]$ if and only if $0 < \mu \leq \bar{\mu}(\gamma)$.

First, since $4\gamma^2 - 2\gamma + 2 = 4(\gamma - \frac{1}{4})^2 + \frac{7}{4} > 0$, we have $\lambda_2 > 0$ for all $\mu > 0$. Next, since $1 + 4\gamma - 4\gamma^2 > 0$ for $\gamma \in (0, 1)$:

$$\lambda_1 \geq 0 \Leftrightarrow \mu \leq \frac{2\gamma(2\gamma + 3)}{1 + 4\gamma - 4\gamma^2} = \bar{\mu}(\gamma).$$

Finally, we have:

$$(1 + 4\gamma - 4\gamma^2) - (6\gamma - 5) = 2(1 - \gamma)(2\gamma + 3) > 0 \quad \text{for } \gamma \in (0, 1).$$

Hence, whenever $\mu \leq \bar{\mu}(\gamma)$, we have:

$$\mu(6\gamma - 5) \leq \mu(1 + 4\gamma - 4\gamma^2) \leq 2\gamma(2\gamma + 3),$$

which implies $\lambda_3 \geq 0$.

By combining these observations, if $0 < \mu \leq \bar{\mu}(\gamma)$, then $\lambda_1, \lambda_2, \lambda_3 \geq 0$, and hence $\lambda \in \Delta^3$. Conversely, if $\mu > \bar{\mu}(\gamma)$, then $\lambda_1 < 0$, and hence $\lambda \notin \Delta^3$. Therefore, $\lambda \in \Delta^3$ if and only if $0 < \mu \leq \bar{\mu}(\gamma)$, and consequently $x^\mu = x^*$ if and only if $0 < \mu \leq \bar{\mu}(\gamma)$. $\qquad \square$

