# OpenReview forum: "Asymmetric Perturbation in Solving Bilinear Saddle-Point Optimization"
_ICML.cc/2026/Conference — ICML 2026 spotlight_

### Official Review · Reviewer_AxD5 · 2026-02-27

**Soundness:** 3
**Presentation:** 2
**Significance:** 3
**Originality:** 3
**Overall Recommendation:** 5
**Confidence:** 3

**Summary:**

This paper studies bilinear saddle point problems and proposes a method of perturbing the players' utilities in an asymmetric fashion. While symmetric perturbations have been widely studied in the literature, it is known that they result in a change to the equilibrium of the game, and often requires carefully designed adaptive perturbation strengths to converge. The main conceptual contribution in this paper is that if the perturbations are performed on one player's payoffs and not the other, the resulting minimax/maximin problem retains the original equilibrium strategy for the player (contingent on perturbation strength). Utilizing this, the authors propose a first order alternating method based on GDA where the x player is updated using the gradient of the perturbed payoff function while the y player updates using the original payoff. Last-iterate convergence with an linear rate is shown for this procedure, and empirically it is observed that it outperforms standard methods like alternating GDA and symmetrically perturbed alternating GDA. The paper also gives a parameter free version of AsymP-GDA, showing that linear convergence to equilibria still holds. Finally, the results are extended to the setting of extensive-form games, and the authors evaluate a variant of their algorithm using a dilated Euclidean regularizer, exhibiting faster last iterate convergence to equilibria than standard methods.

**Compliance With Llm Reviewing Policy:**

Affirmed.

**Final Justification:**

The author rebuttal has addressed my concerns adequately, and so I maintain my support of the paper.

**Key Questions For Authors:**

- In the EFG experiments, it appears that for certain games, dilated OGDA performs comparatively well to AsymP-GDA. Is there any intuition as to why this is the case? Moreover, would it also make sense to consider an optimistic, simultaneous but asymmetrically perturbed GDA algorithm? This is reminiscent of the work of [1] that studies dilated OMD in perturbed EFGs.

[1] Liu, Mingyang, et al. "The power of regularization in solving extensive-form games." arXiv preprint arXiv:2206.09495 (2022).

**Limitations:**

Yes

**Strengths And Weaknesses:**

### Strengths
- The paper introduces a simple yet effective perturbation method that guarantees fast convergence to un-modified equilibria. The primary significance here is the simplicity of the analysis, while also establishing conceptually interesting results that separate the asymmetric and symmetric cases. This resolves several key problems that arise from using perturbations to improve convergence rates in saddle point problems.
- The study of parameter-free AsymP-GDA is a particularly nice contribution, and ties together the main results (Thms 3.1 and Cor 4.2). In particular, it is not clear how useful the parametrized version of AsymP-GDA would be in practice, and a parameter-free method that still converges in linear time is, in my opinion, a significant contribution.

### Weaknesses
- A major weakness for me is the results and analysis in the EFG setting. While the algorithm does appear to perform well in standard benchmarks, I feel that the section lacks a theoretical analysis to complement the empirical results. In particular, it is not clear if the analysis from the standard algorithm extends to the dilated case, due to the asymmetric nature of the update. If this is indeed true, then it would be need to be justified in the main text, and otherwise, some discussion on the challenges that arise would be warranted.
- The experiments for EFGs also utilize a hyperparameter tuning procedure to find suitable values of $\mu$ for the asymmetrically perturbed algorithms. I feel this section is lacking a comparison between the tuned version and the parameter-free version. While of course last-iterate convergence will hold between the two, it would be useful to understand if the parameter-free implementation is actually computationally efficient compared to tuning $\mu$.
- This is a minor comment, but Figure 2 takes up a large amount of space which could probably be used to add the CFR comparisons or even more proof details, which would make the overall contributions clearer to readers.

---

> ### Author Rebuttal · Authors · 2026-03-31
>
> We thank you for your positive feedback and constructive comments. The detailed answers to each of your questions are provided below.
>
> ---
>
> > A major weakness for me is the results and analysis in the EFG setting. While the algorithm does appear to perform well in standard benchmarks, I feel that the section lacks a theoretical analysis to complement the empirical results. In particular, it is not clear if the analysis from the standard algorithm extends to the dilated case, due to the asymmetric nature of the update. If this is indeed true, then it would be need to be justified in the main text, and otherwise, some discussion on the challenges that arise would be warranted.
> >
>
> The proof of Theorem 3.1 does not directly extend to the dilated Euclidean regularizer. The key reason is that, unlike the standard Euclidean regularizer, **the dilated Euclidean regularizer does not enjoy the same kind of global smoothness over the strategy space**, as also noted in Lee et al. (2021). Our proof of Theorem 3.1 relies on the following first-order upper bound:
>
> $$
> \psi(x^{\ast}) - \psi(x) \leq \langle \nabla \psi(x^{\ast}), x^{\ast} - x\rangle \leq C\\|x^{\ast}-x\\|,
> $$
>
> for some global constant $C>0$, which follows from the uniform boundedness of the gradient of the standard Euclidean regularizer over the strategy space. In contrast, in the dilated setting, $\nabla \psi_{\mathrm{dil}}(x)$ depends on inverse parent reach probabilities and may become arbitrarily large near the boundary. Therefore, an analogous global bound is generally unavailable. We will clarify this technical obstacle in the revised manuscript.
>
> ---
>
> > The experiments for EFGs also utilize a hyperparameter tuning procedure to find suitable values of for the asymmetrically perturbed algorithms. I feel this section is lacking a comparison between the tuned version and the parameter-free version. While of course last-iterate convergence will hold between the two, it would be useful to understand if the parameter-free implementation is actually computationally efficient compared to tuning $\mu$.
> >
>
> > This is a minor comment, but Figure 2 takes up a large amount of space which could probably be used to add the CFR comparisons or even more proof details, which would make the overall contributions clearer to readers.
> >
>
> Thank you for your kind suggestion! In the revised manuscript, we will include a direct empirical comparison between the tuned and parameter-free implementations. We will also reconsider the allocation of space for figures and text to better highlight these comparisons and, where space permits, add CFR comparisons and/or further theoretical details.
>
> ---
>
> > In the EFG experiments, it appears that for certain games, dilated OGDA performs comparatively well to AsymP-GDA. Is there any intuition as to why this is the case?
> >
>
> As you pointed out, dilated OGDA is competitive with AsymP-GDA in one of our EFG benchmarks, namely Goofspiel (4 cards). Based on the experimental results, our intuition is that, in this particular game, many information sets require very little learning from the initial strategy. In particular, the equilibrium strategy appears to be close to uniform at many information sets in this instance. Since we use the uniform strategy as the initialization in our EFG experiments, the initial strategy is already close to equilibrium on a large portion of the game tree, leaving little room for further improvement through learning. As a result, the advantage of AsymP-GDA may be less pronounced in this game than in the other EFG benchmarks.
>
> ---
>
> > Moreover, would it also make sense to consider an optimistic, simultaneous but asymmetrically perturbed GDA algorithm? This is reminiscent of the work of [1] that studies dilated OMD in perturbed EFGs.
> >
> > [1] Liu, Mingyang, et al. "The power of regularization in solving extensive-form games." arXiv preprint arXiv:2206.09495 (2022).
> >
>
> Yes, we believe such an extension is indeed meaningful. In this work, however, we intentionally focused on the simplest version of AsymP-GDA in order to isolate the effect of asymmetric perturbation itself. Along this direction, it should also be possible to establish convergence rate guarantees for an optimistic version of AsymP-GDA analogous to those in Theorem 4.1 and Corollary 4.2.

---

> > ### Author Rebuttal · Reviewer_AxD5 · 2026-04-01
> >
> > The comments from the authors have clarified my concern about the analysis extending to the EFG case. It would certainly be an interesting future direction, but as it stands adding a discussion on this would be useful for the present work. Overall I find that the paper still introduces a qualitatively distinct contribution to the community, and so I will maintain my score and support of the paper.

---

> > > ### Author Response · Authors · 2026-04-06
> > >
> > > We sincerely thank the reviewer for the thoughtful follow-up and for the positive assessment of our work. We are glad that our rebuttal helped clarify the relevant points. Thank you again for your time and consideration.

---

### Official Review · Reviewer_1Phw · 2026-03-13

**Soundness:** 3
**Presentation:** 4
**Significance:** 4
**Originality:** 4
**Overall Recommendation:** 5
**Confidence:** 4

**Summary:**

This paper considers the classical setting of constrained min-max optimization of a bilinear objective: $\min_{x \in X} \max_{y \in Y} x^\top A y$ where $X$ and $Y$ are polytopes, and more specifically, the question of designing algorithms that converge in the last-iterate sense. It is well-known that simultaneous and alternating projected gradient descent-ascent (GDA) both fail, while proximal-point-type methods succeed. It is also well-understood that introducing regularization in the form of two terms $+\frac\mu2 \|x\|^2 - \frac\mu2 \|y\|^2$ suffices to make GDA converge, albeit at the cost of a "bias": the limit point of the algorithm can be up to $O(\mu)$-suboptimal (in duality gap, the usual suboptimality metric for this setting). This paper shows that introducing regularization for only one of the variables, e.g., adding a term $+\frac\mu2 \|x\|^2$, is still sufficient to ensure exponential convergence of alternating GDA, and remarkably does not introduce a "bias" as long as $\mu < \alpha/(\sup_X \|x\|)$ for some problem-dependent $\alpha$. Although $\alpha$ can be difficult to estimate a priori, it is shown that a simple modification of the algorithm (using a halving trick) allows to dispense knowing it a priori while conserving the exponential convergence guarantee. The paper also shows how their proposed asymmetrically-perturbed GDA method can be adapted to tackle extensive-form games.

**Compliance With Llm Reviewing Policy:**

Affirmed.

**Final Justification:**

I maintain my positive assessment of this work and my recommendation to accept it. (The rebuttal only helped to clarify some relatively minor questions I had, all of which were adequately answered.)

**Key Questions For Authors:**

1. Is NashConv a standard term? is it the same as Nikaido-Isoda error (this name seems more common)? Also, what does "Conv" stand for?
2. Concerning the parameter $\alpha$ (defined on line 216): There is one $\alpha$ for $x$ and another one for $y$. How different can they be?
3. One run of the proposed algorithm allows to compute $x^\*$. Is there a way to efficiently get $y^\*$ with the knowledge of $x^\*$, that would be more efficient than just reapplying the same method with players roles flipped?
4. What is the constant in the exponent in Corollary 4.2? Is it written out somewhere in the appendix? It would have been nice to see it written, or to have a reference to where it can be found in the appendix.
5. Does the maximal $\mu$ such that the algorithm converges match your theoretical bound of $\alpha/(\sup_X \|x\|)$ in practice?

**Limitations:**

Yes

**Strengths And Weaknesses:**

Overall, this is a well-written paper introducing a remarkable original idea to the field of min-max optimization. While the key underlying phenomenon (the "1-homogeneous stability", or error-bound type inequality, satisfied by polytope-constrained bilinear saddle point problems) has been known for some time, the fact that it can be exploited in the context of explicit regularization (what the authors call payoff perturbation techniques) is a valuable contribution. It is possible that this fact identified by the authors will also have implications for settings.

The structure and the presentation of the paper are quite clear. Just a few points/suggestions:
- lines 31 and 61 and 424 (second column) may be confusing at first reading, because they only make sense if the reader knows that the allowable range of $\mu$ is not easy to estimate a priori.
- line 215, the paper by Wei et al (2021) discusses several such inequalities. It could be worth specifying which part of the paper this citation refers to.
- the caption of Figure 3 should clarify that this is for infinitesimal step-sizes.
- line 328, an earlier relevant reference than Zhang et al (2022) might be Lemma G.2 in Nagarajan and Kolter (2018) ("Gradient descent GAN optimization is locally stable")

The paper is sound as far as I have checked. However I have not checked the proofs for the convergence analysis of the proposed algorithm, nor the experiment details on extensive-form games. So my rating for "Soundness" should be taken with caution.

---

> ### Author Rebuttal · Authors · 2026-03-31
>
> We thank you for your positive feedback and constructive comments. The detailed answers to each of your questions are provided below.
>
> ---
>
> > lines 31 and 61 and 424 (second column) may be confusing at first reading, because they only make sense if the reader knows that the allowable range of $\mu$ is not easy to estimate a priori.
> >
>
> We agree with this comment. To make the motivation for the parameter-free variant clearer, the manuscript should more explicitly state that AsymP-GDA requires choosing $\mu$ within an allowable range that may be difficult to estimate a priori. In the revised manuscript, we will add this limitation of AsymP-GDA in the abstract and conclusion so that the contribution of the parameter-free variant is easier to understand from the outset.
>
> ---
>
> > line 215, the paper by Wei et al (2021) discusses several such inequalities. It could be worth specifying which part of the paper this citation refers to.
> >
>
> > the caption of Figure 3 should clarify that this is for infinitesimal step-sizes.
> >
>
> > line 328, an earlier relevant reference than Zhang et al (2022) might be Lemma G.2 in Nagarajan and Kolter (2018) ("Gradient descent GAN optimization is locally stable")
> >
>
> Thank you for these helpful suggestions. In the revised manuscript, we will improve the presentation regarding these points by making the citation to Wei et al. (2021) more precise, clarifying the caption of Figure 3, and adding the earlier reference by Nagarajan and Kolter (2018).
>
> ---
>
> > Is NashConv a standard term? is it the same as Nikaido-Isoda error (this name seems more common)? Also, what does "Conv" stand for?
> >
>
> Yes, NashConv is standard terminology in the learning-in-games literature, as used for example in Srinivasan et al. (2018) and Timbers et al. (2022), and it is also widely used in frameworks such as OpenSpiel (Lanctot et al., 2019). In our setting, it is equivalent to the Nikaido-Isoda error. Here, “Conv” stands for “convergence.”
>
> * Srinivasan, S., et al. Actor-Critic Policy Optimization in Partially Observable Multiagent Environments. In NeurIPS, 2018.
> * Timbers, F., et al. Approximate Exploitability: Learning a Best Response in Large Games. In IJCAI, 2022.
> * Lanctot, M., et al. OpenSpiel: A Framework for Reinforcement Learning in Games. arXiv:1908.09453, 2019.
>
> ---
>
> > Concerning the parameter $\alpha$ (defined on line 216): There is one for $x$ and another one for $y$. How different can they be?
> >
>
> In general, the corresponding $\alpha$ values for players $x$ and $y$ can be different. Both depend on the underlying game instance through the game matrix $A$, and their relative scales may vary across games. While it is difficult to estimate this difference a priori in general, one point we would like to emphasize is that the parameter-free variant of AsymP-GDA does not require any prior knowledge of these constants.
>
> ---
>
> > Is there a way to efficiently get $y^{\ast}$ with the knowledge of $x^{\ast}$, that would be more efficient than just reapplying the same method with players roles flipped?
> >
>
> Unfortunately, we do not currently know a generic method that is provably more efficient than solving the analogous problem with the player roles flipped. That said, once $x^{\ast}$ has been obtained, using it as the initial strategy for player $x$ when solving for player $y$ may be practically beneficial and could accelerate convergence in practice.
>
> ---
>
> > What is the constant in the exponent in Corollary 4.2? Is it written out somewhere in the appendix?
> >
>
> To clarify, Corollary 4.2 follows directly from Theorem 4.1, so the constant in the exponent is exactly the same as in Theorem 4.1. More specifically, in Appendix F.1, Hoffman’s bound implies that there exists a constant $\beta>0$ such that:
>
> $$
> \\|\Pi_{Z^{\mu}}(z^t) - z^t\\| \leq \beta\\|z^{t+1} - z^t\\|.
> $$
>
> Using this inequality, we obtain the linear convergence rate:
>
> $$
> \\|\Pi_{Z^{\mu}}(z^{t+1}) - z^{t+1}\\|^2 \leq \left(\frac{1}{1+1/(2\beta^2)}\right)^t\\|\Pi_{Z^{\mu}}(z^1) - z^1\\|^2.
> $$
>
> We will add an explicit reference from Corollary 4.2 and Theorem 4.1 to Appendix F.1.
>
> ---
>
> > Does the maximal $\mu$ such that the algorithm converges match your theoretical bound of $\alpha/(\sup_X |x|)$ in practice?
> >
>
> **The upper bound on $\mu$ in Theorem 3.1 is essentially tight**, at least in terms of its dependence on $\alpha$, as can be seen from a simple bilinear game. Consider a bilinear game with $X=[0,1]$, $Y=[1,2]$, and $A=-1$. In this game, the minimax strategy is $x^{\ast}=1$, and the near-linear growth condition used in the proof holds with equality with $\alpha=1$. Moreover, we have $x^{\mu} = x^{\ast}$ for all $\mu \leq \alpha /\sup_{x\in X}\\|x\\|=1$, whereas $x^{\mu}\neq x^{\ast}$ for $\mu>1$. This shows that the threshold $\alpha/\sup_{x\in X}\\|x\\|$ is tight in this example.

---

> > ### Author Rebuttal · Reviewer_1Phw · 2026-04-01
> >
> > Thank you for your response! My questions have been adequately answered (though to be clear, none of them were essential). I maintain my positive appreciation of this paper.

---

> > > ### Author Response · Authors · 2026-04-06
> > >
> > > We sincerely thank the reviewer for the thoughtful follow-up and for the positive assessment of our work. We are glad that our rebuttal helped clarify the relevant points. Thank you again for your time and consideration.

---

### Official Review · Reviewer_bREA · 2026-03-15

**Soundness:** 3
**Presentation:** 3
**Significance:** 2
**Originality:** 3
**Overall Recommendation:** 4
**Confidence:** 4

**Summary:**

This paper proposes asymmetric (one-sided) payoff perturbations for zero-sum games and analyzes first-order methods (e.g., AsymP-GDA) under this perturbation. The key message is an equilibrium invariance regime: for sufficiently small (\mu), the perturbed game can recover an original-game minimax strategy on the perturbed side (e.g., (x^\mu \in X^)). The paper provides convergence guarantees (to a perturbed saddle set) and empirical results on matrix and extensive-form games, arguing improved practicality compared to symmetric perturbations that require budget-dependent tuning.

**Compliance With Llm Reviewing Policy:**

Affirmed.

**Final Justification:**

Thank you for the author's feedback; I will maintain my score.

**Key Questions For Authors:**

1. Clarify the NashConv definition in the fixed-(\mu) plots (original vs perturbed vs one-sided) and adjust labeling or add explanation accordingly.
2. Add/strengthen a paired-NE baseline: compare methods under the goal most relevant in games—achieving low original-game NashConv for a pair ((x,y)), reporting total compute (including any second solve if needed).

**Limitations:**

Please refer to what is described in "Strengths And Weaknesses".

**Strengths And Weaknesses:**

1. Soundness. The theory and algorithms are generally convincing for the stated goal of recovering a single-side minimax/maximin strategy under one-sided perturbation. However, I have one major clarification request: some experiments (e.g., fixed (\mu) curves) appear to report NashConv (\approx 10^{-15}). If this NashConv is computed for the original game using the pair ((x_t,y_t)), then it seems inconsistent with the one-sided nature of the method (since fixed (\mu>0) typically yields (y^\mu\neq y^)). Please explicitly define what is plotted: (i) original-game NashConv of ((x_t,y_t)), (ii) a perturbed gap/objective, or (iii) a one-sided exploitability. If it is not original-game NashConv, I suggest renaming the metric/caption accordingly; if it is, please explain why pairwise NashConv can reach machine precision under fixed (\mu).

2. Presentation. The paper is well written overall, but the above ambiguity is easy for readers to misinterpret. A short clarification near the first relevant figure (or in the caption) would greatly improve clarity: what solution concept is targeted (single-side vs pair) and what exact metric is computed.

3. Significance. The work addresses an important and active area (last-iterate convergence and practical game-solving). That said, many readers ultimately care about the Nash equilibrium pair ((x^,y^)). I recommend adding a baseline/evaluation that reflects this: e.g., “AsymP run twice” (perturb (x) then perturb (y)) or an explicit procedure to output a pair, and compare time/iterations to reach a target original-game NashConv against standard baselines.

4. Originality. The asymmetric perturbation + invariance regime is a nice and nontrivial conceptual angle, and the parameter-free continuation idea makes it practically appealing.

---

> ### Author Rebuttal · Authors · 2026-03-31
>
> We thank you for your positive feedback and constructive comments.
>
> Also, thank you for the important clarification request regarding the NashConv evaluation in the EFG experiments. We agree that the manuscript should define the plotted metric and the experimental protocol much more explicitly. In the EFG experiments, we do not use a single run of AsymP-DGDA to produce the pair $(x^t, y^t)$. Instead, to recover equilibrium strategies for both players in the original game, **we run AsymP-DGDA separately for each player**: one run outputs player $x$'s strategy sequence $\{x^t\}$, and another run (with the player roles flipped) outputs player $y$'s strategy sequence $\{y^t\}$. **We then form the strategy pair $(x^t,y^t)$ by combining these two outputs**, and **the plotted quantity is the original-game NashConv of this combined pair**. In this sense, the paired evaluation suggested in your comment is already the one used in our experiments; however, we agree that this is not stated clearly enough in the current manuscript. We will revise the experimental section and figure captions to make the evaluation protocol, the target solution concept, and the total computation across the two runs explicit.

---

> > ### Author Rebuttal · Reviewer_bREA · 2026-04-01
> >
> > The rebuttal clarifies that the reported original-game NashConv is obtained by combining two separate asymmetric runs, one for each player. This resolves the metric-definition issue. However, the paper itself does not currently make this clear, and Figure 4’s x-axis explanation appears to account only for per-iteration update cost within a run, not the additional cost of a second role-flipped solve needed to form an equilibrium pair. If the plotted NashConv indeed comes from two runs, the total compute should be charged accordingly; otherwise the comparison may favor AsymP unfairly.

---

> > > ### Author Response · Authors · 2026-04-01
> > >
> > > Thank you for this helpful follow-up!
> > >
> > > We agree that, if the reported original-game NashConv is obtained by combining two separate AsymP-DGDA runs, then the costs of both runs must be accounted for in the comparison. **This is indeed how Figure 4 is constructed**. As already stated in the caption of Figure 4, we report the **total number of strategy updates** on the x-axis rather than the number of iterations. For AsymP-DGDA, this total count includes the updates from both runs, namely the run used to compute player $x$'s strategy and the run used to compute player $y$'s strategy. Equivalently, when reporting the paired NashConv for AsymP-DGDA, each run is given only half of the total iteration budget, so that the overall strategy-update cost matches that of the other methods. Therefore, **the comparison in Figure 4 is intended to be fair in terms of total computational cost**. We agree, however, that this point could be stated more clearly, and we will revise the caption and the surrounding text accordingly.

---

### Official Review · Reviewer_8Uzj · 2026-03-15

**Soundness:** 2
**Presentation:** 3
**Significance:** 3
**Originality:** 3
**Overall Recommendation:** 4
**Confidence:** 3

**Summary:**

The paper studies a well-known problem in bilinear zero-sum games, namely the saddle-point problem, which is a fundamental topic in game theory. Due to its importance, there is a substantial body of prior work devoted to developing fast algorithms for computing an approximate, or even exact, Nash equilibrium. One well-known approach (symmetric perturbation) reformulates the problem by introducing strongly convex, concave perturbations of the payoffs, thereby making the problem strongly convex-concave and enabling linear-time convergence (or exponentially fast convergence, in the terminology used in the game theory community) under standard algorithms. However, this paper argues that, despite the favorable convergence properties of symmetric perturbation, it introduces an approximation error that is inherent to the method itself. For this reason, the authors show that an alternative asymmetric perturbation may alleviate this issue while still maintaining the same favorable convergence properties.

**Compliance With Llm Reviewing Policy:**

Affirmed.

**Final Justification:**

The authors have addressed my concern to an almost adequate extent, so I have decided to increase my score accordingly. At this point, however, I am still not certain that the work is at the level required for publication, and I therefore retain some reservations, which explain my weak accept recommendation.

**Key Questions For Authors:**

1. As discussed above, I am unsure whether the proposed method actually resolves the issues that occasionally arise with symmetric perturbation. In other words, I think both methods may suffer from similar weaknesses, and it is not clear to me that asymmetric perturbation truly does what the authors claim. For instance, if we consider

A = np.array([
    [1.0, -1.0,  2.0],
    [0.0,  3.0, -2.0],
    [-1.0, 1.0,  0.0]
])

instead of the game used in Figure 1, then the behavior of the gap appears much more irregular and as well behaved as suggested by Figure 1(b).

2. I do not understand the comparison to Nesterov (2005), but probably I am missing something. That method was originally introduced for the minimization of non-differentiable functions, where the addition of a strongly convex function makes it smooth and thus it yields faster convergence rates. The connection to bilinear saddle-point problems is not clear to me.

3. It would help if the authors could state more explicitly under what conditions asymmetric perturbation provides a genuine practical advantage over symmetric perturbation.

**Limitations:**

The paper does not highlight any limitation or societal impact, but I do not see any specific negative societal impact here.

**Strengths And Weaknesses:**

As I pointed out in the abstract, this work, as the authors claim, identifies a limitation of existing linear-convergence methods, although it is still not entirely clear to me why this claim should hold as broadly as suggested. That said, I have not seen this exact idea elsewhere before, so the idea appears quite original and could motivate interesting further research, particularly regarding when and whether this method is truly preferable to symmetric perturbation. Overall, this work focuses on an important concept.

That being said, I do not find the overall story fully convincing. Starting with Theorem 3.1, I am not sure how informative the result really is. The existence of an upper bound on $\mu$ is more or less expected, even in the case of symmetric perturbation, and it does not necessarily provide meaningful guidance for the design of an algorithm that can exploit this knowledge. For instance, by scaling the values of the payoff matrix, this upper bound also changes. Moreover, this constant can be arbitrarily small, in which case the method essentially reduces to the unperturbed setting. Similarly, Figure 3 is somewhat misleading. As is usually the case with symmetric perturbation methods, the perturbation constant $\mu$ mainly depends on the number of iterations and, as a result, on the approximation one aims to achieve. In contrast, Figure 3 gives the impression that symmetric perturbation will always stay away from equilibrium. Specifically, $\mu$ can also be time-varying, which would eliminate the “issue” here.

Another concern as per Figure 1 (b) is that the behavior of the asymmetric perturbation is bound to that particular game; in other words, it is not a general behavior intrinsic to the (asymmetric perturbation) method itself. For example, I presented a game in the Questions section, for which the experiments show that the metric the authors use, somewhat like the Nash gap, is rather erratic.

Overall, I am not really convinced that this method resolves the shortcomings of symmetric perturbation, and for this reason I believe the authors should clarify much more precisely where the real merit of this approach is.

---

> ### Author Rebuttal · Authors · 2026-03-31
>
> Thank you for taking the time to review our work and for providing your valuable feedback. The detailed answers to each of the questions can be found below.
>
> ---
>
> ### On the Major Contributions of Our Study
>
> We would like to highlight that, to the best of our knowledge, our study provides **the first perturbation-based learning algorithm that achieves a fast linear last-iterate convergence rate**. The key ingredient behind this result is Theorem 3.1. Intuitively, Theorem 3.1 establishes the equilibrium invariance property, meaning that **an equilibrium strategy of the asymmetrically perturbed game exactly coincides with one of the original game** for a fixed range of $\mu$. In this sense, one does not need to decrease $\mu$ as a function of the iteration number in order to recover an equilibrium of the original game.
>
> This is qualitatively different from symmetric perturbation. Even in a small-$\mu$ regime, the corresponding invariance property does not generally hold under symmetric perturbation. In fact, Theorem B.1 and Corollary B.2 show that this invariance can fail for every fixed $\mu>0$, no matter how small it is. As a result, symmetric perturbation typically requires iteration-dependent tuning of $\mu$ to recover the original equilibrium asymptotically, and the best currently available guarantees in this direction are slower, at best $\tilde{\mathcal{O}}(1/t)$ (Liu et al., 2023). By contrast, the equilibrium invariance property in Theorem 3.1 is precisely what enables AsymP-GDA to achieve a linear last-iterate convergence rate with a constant $\mu$.
>
> ---
>
> ### On the Practical Advantage of Asymmetric Perturbation
>
> Beyond the conceptual significance of Theorem 3.1, it also leads to a concrete practical advantage. Because asymmetric perturbation admits an iteration-independent range of $\mu$ values for which equilibrium invariance holds, **our method can use a constant perturbation strength and still converge to an equilibrium of the original game at a linear rate**. This is faster than the best currently available guarantees under symmetric perturbation in the same setting. Moreover, by further exploiting this property, **we can construct a parameter-free variant that achieves the same linear rate without requiring prior knowledge of game-dependent constants**.
>
> ---
>
> ### On the General Behavior beyond Figure 1(b)
>
> We would like to emphasize that **the equilibrium invariance behavior under asymmetric perturbation is not specific to the game shown in Figure 1(b)**. Rather, Theorem 3.1 establishes a general theoretical property for the class of bilinear games considered in this paper, under the stated condition on $\mu$. In this sense, Figure 1(b) is intended only as an illustration of the theorem, not as evidence of a game-specific phenomenon. This theoretical property also applies to the game matrix you provided, $A=[ [1, -1, 2], [0, 3, -2], [-1, 1, 0] ]$, and we numerically confirmed that the original equilibrium strategy is recovered for a broad range of $\mu$, specifically for $\mu \leq 15$.
>
> Moreover, our claim is not that the gap as a function of $\mu$ must behave non-erratically. Rather, Theorem 3.1 shows that there exists a range of $\mu$ values for which the original equilibrium strategy is recovered exactly. Outside this range, or when the metric is examined over a broad range of $\mu$, the curve may look more irregular depending on the game instance. This does not contradict the theorem, nor does it suggest that asymmetric perturbation fails to preserve equilibrium invariance within the guaranteed range.
>
> ---
>
> ### On the Comparison to Nesterov (2005)
>
> Thank you for pointing this out. We referred to Nesterov (2005) because that work also considers bilinear saddle-point problems as an application of nonsmooth optimization; see Section 4.2 in Nesterov (2005). There is also a structural similarity in the approach: both methods add a strongly convex term to only one player's payoff function. More specifically, Nesterov’s approach perturbs only the opponent’s payoff, whereas our method perturbs only the payoff of the player being optimized. That said, the purpose is different in our setting: we use this asymmetry to obtain an equilibrium invariance property. We will revise this discussion to make both the similarity and the distinction clearer.

---

> > ### Author Rebuttal · Reviewer_8Uzj · 2026-04-03
> >
> > Thank the authors for the thorough response. Although I feel more confident regarding the comparison to the well-known symmetric perturbation methods, I still believe that Theorem 3.1, which, as the authors note, constitutes the core of the whole approach, is more something to be expected than something particularly informative, since this constant may be arbitrarily small, in which case the method effectively reduces to an unperturbed game. However, I have chosen to increase my score.

---

> > > ### Author Response · Authors · 2026-04-06
> > >
> > > We greatly appreciate the reviewer's recognition of the distinction from symmetric perturbation methods and the decision to increase the score. We also understand the reviewer’s perspective on Theorem 3.1 and will revise the presentation to more clearly explain its connection to the motivation for the parameter-free variant.

---

### Decision · Program_Chairs · 2026-04-30

**Decision:**

Accept (spotlight)

**Comment:**

A common approach in solving zero-sum games is symmetric regularization/perturbation in which the regularization strength is decayed towards zero. This work shows if one employs asymmetric perturbation, the strength can be set to a strictly positive threshold and can still achieve linear convergence to the original equilibrium. All reviewers found this result to be qualitatively new and interesting. Some concerns were raised regarding clarifying the limitations of the proposed approach: a) must run twice to compute both player’s NE strategies, b) parameter threshold could be arbitrarily small, c) Theorem 3.1 doesn’t extend to dilated regularization in EFGs, d) parameter is decayed in a doubling trick although the approach then looks similar to the symmetric approach to some extent, especially if parameter threshold is small. The authors should make sure to explicitly discuss these in their revision. I would also advise the authors to plot results of empirically obtained upper bounds on a small class of instances (e.g., 2x2 matrix games) so a reader gets an idea how often b) is a potential issue in practice and whether some relationship exists between $\alpha$ and e.g., the entropy of $x^*$. Overall, this represents a significant contribution to our understanding of learning in games.

Note: Fig 2, right equation in red should be evaluated at x’ rather than x presumably?